# SYMDIFF: EQUIVARIANT DIFFUSION VIA STOCHASTIC SYMMETRISATION

**Leo Zhang  Kianoosh Ashouritaklimi  Yee Whye Teh  Rob Cornish**
Department of Statistics, University of Oxford

## ABSTRACT

We propose SYMDIFF, a method for constructing equivariant diffusion models using the framework of stochastic symmetrisation. SYMDIFF resembles a learned data augmentation that is deployed at sampling time, and is lightweight, computationally efficient, and easy to implement on top of arbitrary off-the-shelf models. In contrast to previous work, SYMDIFF typically does not require any neural network components that are intrinsically equivariant, avoiding the need for complex parameterisations or the use of higher-order geometric features. Instead, our method can leverage highly scalable modern architectures as drop-in replacements for these more constrained alternatives. We show that this additional flexibility yields significant empirical benefit for $E(3)$-equivariant molecular generation. To the best of our knowledge, this is the first application of symmetrisation to generative modelling, suggesting its potential in this domain more generally.

## 1 INTRODUCTION

For geometrically structured data such as $N$-body systems of molecules or proteins, it is often of interest to obtain a diffusion model that is *equivariant* with respect to some group actions (Xu et al., 2022; Hoogeboom et al., 2022; Watson et al., 2023; Yim et al., 2023). By accounting for the large number of permutation and rotational symmetries within these systems, equivariant diffusion models aim to improve data efficiency (Batzner et al., 2022) and generalisation (Elesedy & Zaidi, 2021). In previous work, equivariant diffusion models have been implemented using *intrinsically equivariant* neural networks, each of whose linear and nonlinear layers are constrained individually to be equivariant, so that the overall network is also equivariant as a result (Bronstein et al., 2021).

There is a growing literature on the practical limitations of intrinsically equivariant neural networks (Wang et al., 2024; Canez et al., 2024; Abramson et al., 2024; Pertigkiozoglou et al., 2024). It has been observed that these models can suffer from degraded training dynamics due to their imposition of architectural constraints, as well as increased computational cost and implementation complexity (Duval et al., 2023a). To address these issues, there has been recent interest in *symmetrisation* techniques for obtaining equivariance instead (Murphy et al., 2018; Puny et al., 2021; Duval et al., 2023b; Kim et al., 2023; Kaba et al., 2023; Mondal et al., 2023; Dym et al., 2024; Gelberg et al., 2024). These approaches offer a mechanism for constructing equivariant neural networks using subcomponents that are not equivariant, which previous work has shown leads to better performing models (Yarotsky, 2022; Kim et al., 2023). However, the advantages of symmetrisation-based equivariance have not yet been explored for generative modelling. This is possibly in part because previous work has solely focused on deterministic equivariance, rather than the more complex condition of *stochastic* equivariance that is required in the context of generative models.

In this paper, we introduce SYMDIFF, a novel methodology for obtaining equivariant diffusion models through symmetrisation, rather than intrinsic equivariance. We build on the recent framework of *stochastic symmetrisation* developed by Cornish (2024) using the theory of Markov categories (Fritz, 2020). Unlike previous work on symmetrisation, which operates on deterministic functions, stochastic symmetrisation can be applied to *Markov kernels* directly in distribution space. We show that this leads naturally to a more flexible approach for constructing equivariant diffusion models than is possible using intrinsic architectures. We apply this concretely to obtain an $E(3)$-equivariant diffusion architecture for modelling $N$-body systems, where $E(3)$ denotes the Euclidean group. For this task, we formulate a custom reverse process that is allowed to be non-Gaussian, for which we derive a

tractable optimisation objective. Our model is stochastically E(3)-equivariant overall without needing any intrinsically E(3)-equivariant neural networks as subcomponents. We also sketch how to extend SYMDIFF to score and flow-based generative models (Song et al., 2020; Lipman et al., 2022).

To validate our framework, we implemented SYMDIFF for de novo molecular generation, and evaluated it as a drop-in replacement for the E(3)-equivariant diffusion of Hoogeboom et al. (2022), which relies on intrinsically equivariant neural networks. In contrast, our model is able to leverage highly scalable off-the-shelf architectures such as Diffusion Transformers (Peebles & Xie, 2023) for all of its subcomponents. We demonstrate this leads to significantly improved empirical performance for both the QM9 and GEOM-Drugs datasets.

## 2 BACKGROUND

We provide here an overview of the underlying theory behind equivariant diffusion modelling. This theory is most conveniently developed in terms of *Markov kernels*, whose definition we recall first.

### 2.1 EQUIVARIANT MARKOV KERNELS

**Markov kernels** At a high level, a *Markov kernel* $k : \mathcal{X} \to \mathcal{Y}$ may be thought of as a *conditional distribution* or *stochastic map* that, when given an input $\mathbf{x} \in \mathcal{X}$, produces a random output in $\mathcal{Y}$ with distribution $k(d\mathbf{y}|\mathbf{x})$. For example, given a function $f : \mathcal{X} \times \mathcal{E} \to \mathcal{Y}$ and a random element $\eta$ of $\mathcal{E}$, there is a Markov kernel $k : \mathcal{X} \to \mathcal{Y}$ for which $k(d\mathbf{y}|\mathbf{x})$ is the distribution of $f(\mathbf{x}, \eta)$[1]. As a special case, every deterministic function $f : \mathcal{X} \to \mathcal{Y}$ may be thought of as a Markov kernel $\mathcal{X} \to \mathcal{Y}$ also. When $k(d\mathbf{y}|\mathbf{x})$ has a *density* (or *likelihood*), we will denote this by $k(\mathbf{y}|\mathbf{x})$, although we note that we can still reason about Markov kernels even when they do not admit a likelihood in this sense.

**Stochastic equivariance** Let $\mathcal{G}$ be a group acting on spaces $\mathcal{X}$ and $\mathcal{Y}$. We will denote this using "dot" notation, so that the action on $\mathcal{X}$ is a function $(g, \mathbf{x}) \mapsto g \cdot \mathbf{x}$. Recall that a function $f : \mathcal{X} \to \mathcal{Y}$ is then *equivariant* if

$$f(g \cdot \mathbf{x}) = g \cdot f(\mathbf{x}) \qquad \text{for all } \mathbf{x} \in \mathcal{X} \text{ and } g \in \mathcal{G}. \tag{1}$$

Here $f$ is purely deterministic, and so this concept must be generalised in order to encompass models whose outputs are stochastic (Bloem-Reddy & Teh, 2020). To this end, Cornish (2024) uses a notion defined for Markov kernels: a Markov kernel $k : \mathcal{X} \to \mathcal{Y}$ is *stochastically equivariant* (or simply *equivariant*) if

$$k(d\mathbf{y}|g \cdot \mathbf{x}) = g \cdot k(d\mathbf{y}|\mathbf{x}) \qquad \text{for all } \mathbf{x} \in \mathcal{X} \text{ and } g \in \mathcal{G}, \tag{2}$$

where the right-hand side denotes the distribution of $g \cdot \mathbf{y}$ when $\mathbf{y} \sim k(d\mathbf{y}|\mathbf{x})$, or in other words the *pushforward* of $k(d\mathbf{y}|\mathbf{x})$ under $g$. When $k$ is obtained from $f$ and $\eta$ as above, equation 2 holds iff

$$f(g \cdot \mathbf{x}, \eta) \stackrel{\mathrm{d}}{=} g \cdot f(\mathbf{x}, \eta) \qquad \text{for all } \mathbf{x} \in \mathcal{X} \text{ and } g \in \mathcal{G},$$

where $\stackrel{\mathrm{d}}{=}$ denotes equality in distribution[2]. If $\eta$ is constant, this says $f$ is deterministically equivariant in the usual sense. Likewise, when $k$ has a conditional density, say $k(\mathbf{y}|\mathbf{x})$, equation 2 holds if

$$k(g \cdot \mathbf{y}|g \cdot \mathbf{x}) = k(\mathbf{y}|\mathbf{x}) \qquad \text{for all } \mathbf{x} \in \mathcal{X}, \mathbf{y} \in \mathcal{Y}, \text{ and } g \in \mathcal{G},$$

provided the action of $\mathcal{G}$ on $\mathcal{Y}$ has unit Jacobian (Cornish, 2024, Proposition 3.18), as will be the case for all the actions we consider. This latter condition recovers the usual formulation of stochastic equivariance considered in the diffusion literature by e.g. Xu et al. (2022); Hoogeboom et al. (2022).

**Stochastic invariance** When the action of $\mathcal{G}$ on $\mathcal{Y}$ is trivial, the condition in equation 1 is referred to as *invariance*. This same idea carries over to Markov kernels also: we say that $k$ above is *stochastically invariant* (or simply *invariant*) if $k(d\mathbf{y}|g \cdot \mathbf{x}) = k(d\mathbf{y}|\mathbf{x})$ for all $g \in \mathcal{G}$ and $\mathbf{x} \in \mathcal{X}$. Importantly, this differs from another natural notion of invariance that also arises in the stochastic context: we will say that a distribution $p(d\mathbf{y})$ on $\mathcal{Y}$ is *distributionally invariant* if it holds that $g \cdot p(d\mathbf{y}) = p(d\mathbf{y})$ for all $g \in \mathcal{G}$. Given a density $p(\mathbf{y})$, this holds equivalently if

$$p(g \cdot \mathbf{y}) = p(\mathbf{y}) \qquad \text{for all } g \in \mathcal{G},$$

again assuming the action on $\mathcal{Y}$ has unit Jacobian.

---

[1] For "nice" choices of $\mathcal{Y}$, the converse also holds by *noise outsourcing* (Kallenberg, 2002, Lemma 3.22).

[2] Note this is more general than the *almost sure* condition from equation (17) of Bloem-Reddy & Teh (2020).

## 2.2 EQUIVARIANT DIFFUSION MODELS

**Denoising diffusion models**   Diffusion models construct a generative model $p_\theta(\mathbf{z}_0)$ of an unknown data distribution $p_{\text{data}}(\mathbf{z}_0)$ on a space $\mathcal{Z}$ by learning to reverse an iterative forward noising process $\mathbf{z}_t$. Following the notation of Kingma et al. (2021), the distribution of $\mathbf{z}_t$ is defined by $q(\mathbf{z}_t|\mathbf{z}_0) = \mathcal{N}(\mathbf{z}_t; \alpha_t \mathbf{z}_0, \sigma_t^2 \mathbf{I})$ for some noise schedule $\alpha_t, \sigma_t > 0$ such that the signal-to-noise ration $\text{SNR}(t) \coloneqq \alpha_t^2/\sigma_t^2$ is strictly monotonically decreasing. The joint distribution of the forward and reverse processes respectively then have the forms:

$$q(\mathbf{z}_{0:T}) = q(\mathbf{z}_0)\prod_{t=1}^{T} q(\mathbf{z}_t|\mathbf{z}_{t-1}) \qquad p_\theta(\mathbf{z}_{0:T}) = p(\mathbf{z}_T)\prod_{t=1}^{T} p_\theta(\mathbf{z}_{t-1}|\mathbf{z}_t) \tag{3}$$

with $q(\mathbf{z}_0) \coloneqq p_{\text{data}}(\mathbf{z}_0)$, and $q(\mathbf{z}_t|\mathbf{z}_{t-1}) = \mathcal{N}(\mathbf{z}_t; \alpha_{t|t-1}\mathbf{z}_{t-1}, \sigma_{t|t-1}^2 \mathbf{I})$, with constants defined as $\alpha_{t|t-1} \coloneqq \alpha_t/\alpha_{t-1}$ and $\sigma_{t|t-1}^2 \coloneqq \sigma_t^2 - \alpha_{t|t-1}^2 \sigma_{t-1}^2$. We take $p(\mathbf{z}_T)$ to also be Gaussian. The reverse process is then trained to maximise the ELBO of $\log p_\theta(\mathbf{z}_0)$, which can be obtained as follows (Sohl-Dickstein et al., 2015):

$$\log p_\theta(\mathbf{z}_0) \geq \mathbb{E}_{q(\mathbf{z}_1|\mathbf{z}_0)}[\log p_\theta(\mathbf{z}_0|\mathbf{z}_1)] - D_{\text{KL}}(q(\mathbf{z}_T|\mathbf{z}_0)||p(\mathbf{z}_T))$$
$$- \sum_{t=2}^{T} \mathbb{E}_{q(\mathbf{z}_t|\mathbf{z}_0)}[D_{\text{KL}}(q(\mathbf{z}_{t-1}|\mathbf{z}_t,\mathbf{z}_0)||p_\theta(\mathbf{z}_{t-1}|\mathbf{z}_t))]. \tag{4}$$

This objective can be efficiently optimised when the reverse process is Gaussian. This is due to the fact that the posterior distributions $q(\mathbf{z}_{t-1}|\mathbf{z}_t, \mathbf{z}_0) = \mathcal{N}(\mathbf{z}_{t-1}; \mu_q(\mathbf{z}_t, \mathbf{z}_0), \sigma_q^2(t)\mathbf{I})$ are Gaussian, and that the KL divergence between Gaussians can be expressed in closed form. To match these posteriors, the reverse process is then typically defined in terms of a neural network $\mu_\theta : \mathcal{Z} \to \mathcal{Z}$ as

$$p_\theta(\mathbf{z}_{t-1}|\mathbf{z}_t) \coloneqq \mathcal{N}(\mathbf{z}_{t-1}; \mu_\theta(\mathbf{z}_t), \sigma_q^2(t)\mathbf{I}). \tag{5}$$

**Invariant and equivariant diffusion**   It is often desirable for $p_\theta(\mathbf{z}_0)$ to be distributionally invariant with respect to the action of a group $\mathcal{G}$. Intuitively, this says that the density $p_\theta(\mathbf{z}_0)$ is constant on the orbits of this action. For the model in equation 3, distributional invariance follows if $p_\theta(\mathbf{z}_T)$ is itself distributionally invariant, and if each $p_\theta(\mathbf{z}_{t-1}|\mathbf{z}_t)$ is stochastically $\mathcal{G}$-equivariant (Xu et al., 2022). All previous work we are aware of has approached this by obtaining a deterministically equivariant $\mu_\theta$, which then implies that the reverse process is stochastically equivariant (Le et al., 2023).

## 2.3 N-BODY SYSTEMS AND $E(3)$-EQUIVARIANT DIFFUSION

Equivariant diffusion models are often applied to model $N$-body systems such as molecules and proteins (Xu et al., 2022; Hoogeboom et al., 2022; Yim et al., 2023). This is motivated by the large number of symmetries present in these system. For example, intuitively speaking, neither the coordinate system nor the ordering of bodies in the system should matter for sampling. We describe the standard components of such models below.

*N*-**body data**   In 3-dimensions, the state of an $N$-body system can be encoded as a pair $\mathbf{z} = [\mathbf{x}, \mathbf{h}] \in \mathbb{R}^{N \times (3+d)}$, where $\mathbf{x} \equiv (\mathbf{x}^{(1)}, \dots, \mathbf{x}^{(N)}) \in \mathbb{R}^{N \times 3}$ describes a set of $N$ points in 3D space, and $\mathbf{h} \equiv (\mathbf{h}^{(1)}, \dots, \mathbf{h}^{(N)}) \in \mathbb{R}^{N \times d}$ describes a set of $N$ feature vectors of dimension $d$. Each feature vector $\mathbf{h}^{(i)}$ is associated with the point $\mathbf{x}^{(i)}$. For example, Hoogeboom et al. (2022) encodes molecules in this way, where each $\mathbf{x}^{(i)}$ denotes the location of an atom, and $\mathbf{h}^{(i)}$ some corresponding properties such as atom type (represented as continuous quantities).

**Center of mass free space**   Intuitively speaking, for many applications, the location of an $N$-body system in space should not matter. For this reason, instead of defining a diffusion on the full space of $N$-body systems directly, previous work (Garcia Satorras et al., 2021a; Xu et al., 2022; Hoogeboom et al., 2022) has set $\mathcal{Z} \coloneqq \mathcal{U}$, where $\mathcal{U}$ is the "center of mass (CoM) free" linear subspace of $\mathbb{R}^{N \times (3+d)}$ consisting of $[\mathbf{x}, \mathbf{h}]$ such that $\bar{\mathbf{x}} \coloneqq \frac{1}{N}\sum_{i=1}^{N} \mathbf{x}^{(i)} = 0$. In this way, samples from their model are always guaranteed to be centered at the origin.

**CoM-free diffusions** To construct their forward and reverse processes to now live entirely on $\mathcal{U}$ instead of $\mathbb{R}^{N \times (3+d)}$, Xu et al. (2022) defines the *projected Gaussian distribution* $\mathcal{N}_{\mathcal{U}}(\mu, \sigma^2 \mathbf{I})$, for $\mu \in \mathcal{U}$ and $\sigma^2 > 0$, as the distribution of $\mathbf{z}$ obtained via the following process:

$$\epsilon \sim \mathcal{N}(0, \mathbf{I}) \qquad \mathbf{z} := \mu + \sigma \operatorname{proj}_{\mathcal{U}}(\epsilon),$$

where $\operatorname{proj}_{\mathcal{U}}$ centers its input in $\mathbb{R}^{N \times (3+d)}$ at the origin, i.e. $\operatorname{proj}_{\mathcal{U}}([\mathbf{x}, \mathbf{h}]) := [\mathbf{x} - (\bar{\mathbf{x}}, \ldots, \bar{\mathbf{x}}), \mathbf{h}]$. By construction, the projected Gaussian distribution is then supported on the linear subspace $\mathcal{U}$. Xu et al. (2022) shows that this distribution has a density with Gaussian form $\mathcal{N}_{\mathcal{U}}(\mathbf{z}; \mu, \sigma^2 \mathbf{I}) \propto \mathcal{N}(\mathbf{z}; \mu, \sigma^2 \mathbf{I})$ defined for all $\mathbf{z}$ living in the subspace $\mathcal{U}$. This allows defining forward and reverse processes $q(\mathbf{z}_{t-1} | \mathbf{z}_t)$ and $p_\theta(\mathbf{z}_{t-1} | \mathbf{z}_t)$ with exactly the same form as in Section 2.2 before, but with $\mathcal{N}_{\mathcal{U}}$ used everywhere in place of $\mathcal{N}$. Since these processes are still Gaussian (albeit on a linear subspace), the KL terms in equation 4 remain tractable, which allows optimising the ELBO in the usual way. This approach does require $\mu_\theta : \mathcal{U} \to \mathcal{U}$ now to be constrained to produce outputs in the subspace $\mathcal{U}$. Prior work has achieved this simply by taking $\mu_\theta$ to be a neural network with $\operatorname{proj}_{\mathcal{U}}$ as its final layer.

**Invariance and equivariance** Intuitively, the ordering of the $N$ points and the orientation of the overall system in 3D space should not matter. To formalise this, let $S_N$ denote the symmetric group of permutations of the integers $\{1, \ldots, N\}$, and $\mathrm{O}(3)$ denote the group of orthogonal $3 \times 3$ matrices. Their product $S_N \times \mathrm{O}(3)$ acts on $N$-body systems by reordering and orthogonally transforming points as follows:

$$(\sigma, R) \cdot [\mathbf{x}, \mathbf{h}] := [R\mathbf{x}^{(\sigma(1))}, \ldots, R\mathbf{x}^{(\sigma(N))}, \mathbf{h}^{(\sigma(1))}, \ldots, \mathbf{h}^{(\sigma(N))}], \qquad (6)$$

where $\sigma \in S_N$ and $R \in \mathrm{O}(3)$. Previous work (Xu et al., 2022; Hoogeboom et al., 2022) has then chosen the model $p_\theta(\mathbf{z}_0)$ to be distributionally invariant to this action. They enforce this via the approach described in Section 2.2, by ensuring that $\mu_\theta$ is deterministically $(S_N \times \mathrm{O}(3))$-equivariant, which implies that the reverse process is stochastically equivariant also.

Following standard terminology in the literature (Garcia Satorras et al., 2021b; Xu et al., 2022; Hoogeboom et al., 2022), we refer to a $(S_N \times \mathrm{O}(3))$-equivariant diffusion defined on the CoM-free space $\mathcal{U}$ as an E(3)-*equivariant diffusion*.

## 3 EQUIVARIANT DIFFUSION VIA STOCHASTIC SYMMETRISATION

In this section, we introduce SYMDIFF and apply it to the problem of obtaining E(3)-equivariant diffusion models for $N$-body systems. We also discuss extensions to score and flow-based generative models (Song et al., 2020; Lipman et al., 2022) in Appendix E.

### 3.1 STOCHASTIC SYMMETRISATION

Recently, Cornish (2024) gave a general theory of neural network *symmetrisation* in the framework of *Markov categories* (Fritz, 2020), encompassing earlier approaches to symmetrisation based on averaging or canonicalisation (Murphy et al., 2018; Puny et al., 2021; Kaba et al., 2023; Kim et al., 2023). This theory applies flexibly and compositionally to general groups and actions, including in the non-compact case, and extends to provide a methodology for symmetrising Markov kernels, which had not previously been considered.

In this work, we will make use of a special case of Example 6.3 of Cornish (2024), which we state now before providing intuition. We denote by $\mathcal{H} \times \mathcal{G}$ the *direct product* of groups $\mathcal{G}$ and $\mathcal{H}$. Recall that an action of $\mathcal{H} \times \mathcal{G}$ on a space $\mathcal{X}$ induces actions of both $\mathcal{H}$ and $\mathcal{G}$ on $\mathcal{X}$ also. For example, $\mathcal{H}$ acts via $h \cdot \mathbf{x} := (h, e_{\mathcal{G}}) \cdot \mathbf{x}$, where $e_{\mathcal{G}}$ is the identity element of $\mathcal{G}$. We will also say that a Markov kernel $\gamma : \mathcal{X} \to \mathcal{G}$ is an *equivariant base case* if it is both $\mathcal{H}$-invariant and $\mathcal{G}$-equivariant, where $\mathcal{G}$ acts on the output space of $\gamma$ by left multiplication, i.e. $g' \cdot g := g'g$. We then have the following result (see Appendix A.1 for a proof).

**Theorem 1.** *Suppose $\mathcal{H} \times \mathcal{G}$ acts on $\mathcal{X}$ and $\mathcal{Y}$, and $\gamma : \mathcal{X} \to \mathcal{G}$ is an equivariant base case. Then every Markov kernel $k : \mathcal{X} \to \mathcal{Y}$ that is equivariant with respect to the induced action of $\mathcal{H}$ gives rise to a Markov kernel $\operatorname{sym}_\gamma(k) : \mathcal{X} \to \mathcal{Y}$ that is equivariant with respect to $\mathcal{H} \times \mathcal{G}$, where $\operatorname{sym}_\gamma(k)(d\mathbf{y} | \mathbf{x})$ may be sampled from as follows:*

$$g \sim \gamma(dg | \mathbf{x}) \qquad \mathbf{y} \sim k(d\mathbf{y} | g^{-1} \cdot \mathbf{x}) \qquad \text{return } g \cdot \mathbf{y}.$$

Intuitively, this result allows us to start with a Markov kernel that is equivariant only with respect to $\mathcal{H}$, and then "upgrade" it to become equivariant with respect to both $\mathcal{H}$ *and* $\mathcal{G}$. As a special case, if $\mathcal{H}$ is the trivial group, then every Markov kernel $k : \mathcal{X} \to \mathcal{Y}$ is $\mathcal{H}$-equivariant. Moreover, in this case $\mathcal{H} \times \mathcal{G} \cong \mathcal{G}$, and so Theorem 1 gives a procedure for obtaining $\mathcal{G}$-equivariant Markov kernels from ones that are completely unconstrained. However, as our $N$-body example will illustrate, it can often be useful to symmetrise a Markov kernel that is already "partially equivariant", which motivates keeping $\mathcal{H}$ general here.

Beyond the existence of an equivariant base case, Theorem 1 is completely generic and requires no assumptions on the groups and actions involved. As explained in Section 5.1 of Cornish (2024), this is also the only natural procedure that can be defined in this way without further assumptions.

**Recursive symmetrisation**   The symmetrisation procedure defined by Theorem 1 requires $\gamma$ already to satisfy two equivariance constraints. In effect, this pushes back the problem of obtaining $(\mathcal{H} \times \mathcal{G})$-equivariant Markov kernels to the choice of $\gamma$, which mirrors the situation in the deterministic setting also (Puny et al., 2021; Kim et al., 2023). Whenever $\mathcal{G}$ is compact, Example 6.3 of Cornish (2024) gives a suitable choice as $\gamma(dg|\mathbf{x}) \coloneqq \lambda(dg)$ , where $\lambda$ denotes the *Haar measure* on $\mathcal{G}$ (Kallenberg, 1997). Other choices could also be made here on a case-by-case basis, such as using intrinsically equivariant neural networks if desired. To obtain greater modelling flexibility, Cornish (2024) also proposes a *recursive* approach to obtaining $\gamma$. Specifically, the idea is to set

$$\gamma \coloneqq \mathsf{sym}_{\gamma_0}(\gamma_1) \tag{7}$$

where $\gamma_0, \gamma_1 : \mathcal{X} \to \mathcal{G}$ are Markov kernels, and $\gamma_0$ is an equivariant base case (e.g. the Haar measure), but where now $\gamma_1$ is only required to be $\mathcal{H}$-invariant, and may behave arbitrarily with respect to $\mathcal{G}$. We note this recursive approach exploits the stochastic nature of the procedure in Theorem 1 and would not be possible using deterministic symmetrisation methods here instead.

## 3.2   SYMDIFF: SYMMETRISED DIFFUSION

We propose to use stochastic symmetrisation to obtain a diffusion process as in Section 2.2 whose reverse kernels are stochastically equivariant. Specifically, suppose some product group $\mathcal{H} \times \mathcal{G}$ acts on our state space $\mathcal{Z}$. For each timestep $t \in \{1, \ldots, T\}$, we will choose some $\mathcal{H}$-equivariant Markov kernel $k_\theta : \mathcal{Z} \to \mathcal{Z}$ that admits a conditional density $k_\theta(\mathbf{z}_{t-1}|\mathbf{z}_t)$. Similarly, we will choose some Markov kernel $\gamma_\theta : \mathcal{Z} \to \mathcal{G}$ that satisfies the conditions of Theorem 1 when $\mathcal{X} = \mathcal{Z}$. With these components, we will define our equivariant reverse process to be

$$p_\theta(\mathbf{z}_{t-1}|\mathbf{z}_t) \coloneqq \mathsf{sym}_{\gamma_\theta}(k_\theta)(\mathbf{z}_{t-1}|\mathbf{z}_t), \tag{8}$$

which is guaranteed to be $(\mathcal{H} \times \mathcal{G})$-equivariant by Theorem 1. This defines a conditional *density*, not just a Markov kernel, as a consequence of the next result. For the proof, see Appendix A.2.

**Proposition 1.** *Assume the same setup as Theorem 1, and for each fixed $g \in \mathcal{G}$, let $k(d\mathbf{y}|g, \mathbf{x})$ be the distribution of the following generative process:*

$$\mathbf{y} \sim k(d\mathbf{y}|g^{-1} \cdot \mathbf{x}) \qquad return\ g \cdot \mathbf{y}.$$

*If $k(d\mathbf{y}|\mathbf{x})$ has a density $k(\mathbf{y}|\mathbf{x})$, then $k(d\mathbf{y}|g, \mathbf{x})$ has a density $k(\mathbf{y}|g, \mathbf{x})$, and $\mathsf{sym}_\gamma(k)(d\mathbf{y}|\mathbf{x})$ has*

$$\mathsf{sym}_\gamma(k)(\mathbf{y}|\mathbf{x}) = \mathbb{E}_{\gamma(dg|\mathbf{x})}[k(\mathbf{y}|g, \mathbf{x})]$$

*as a density. If the action on $\mathcal{Y}$ has unit Jacobian, then we may write $k(\mathbf{y}|g, \mathbf{x}) = k(g^{-1} \cdot \mathbf{y}|g^{-1} \cdot \mathbf{x})$.*

**Training objective**   We would like to learn the parameters $\theta$ using the ELBO from equation 4. However, in general, we do not have access to the densities $p_\theta(\mathbf{z}_{t-1}|\mathbf{z}_t)$ from equation 8 in closed form, since this requires computing the expectation as in Proposition 1. As such, we cannot compute the ELBO directly. However, since $\log$ is concave, Jensen's inequality allows us to bound

$$\log p_\theta(\mathbf{z}_{t-1}|\mathbf{z}_t) \geq \mathbb{E}_{\gamma_\theta(dg|\mathbf{z}_t)}[\log k_\theta(\mathbf{z}_{t-1}|g, \mathbf{z}_t)].$$

Since the ELBO in equation 4 depends linearly on $\log p_\theta(\mathbf{z}_{t-1}|\mathbf{z}_t)$, this allows us to also bound

$$\log p_\theta(\mathbf{z}_0) \geq \overbrace{\mathbb{E}_{q(\mathbf{z}_1|\mathbf{z}_0), \gamma_\theta(dg|\mathbf{z}_1)}[\log k_\theta(\mathbf{z}_0|g, \mathbf{z}_1)]}^{-\mathcal{L}_1} - D_{\mathrm{KL}}(q(\mathbf{z}_T|\mathbf{z}_0)||p(\mathbf{z}_T))$$
$$- \sum_{t=2}^{T} \overbrace{\mathbb{E}_{q(\mathbf{z}_t|\mathbf{z}_0), \gamma_\theta(dg|\mathbf{z}_t)}[D_{\mathrm{KL}}(q(\mathbf{z}_{t-1}|\mathbf{z}_t, \mathbf{z}_0)||k_\theta(\mathbf{z}_{t-1}|g, \mathbf{z}_t))]}^{\mathcal{L}_t}, \tag{9}$$

where the right-hand side is a tractable lower bound to the original ELBO in equation 4. We take this new bound as our objective used to train our SYMDIFF model. By a similar argument as in Remark 7.1 of Cornish (2024), if the model is sufficiently expressive, then optimising this new bound is equivalent to optimising the original ELBO (see Appendix D).

**Comparison with deterministic symmetrisation** An alternative approach to equation 8 would be to obtain $\mu_\theta$ in equation 5 via *deterministic* symmetrisation. However, many deterministic techniques (Puny et al., 2021; Kim et al., 2023) involve a Monte Carlo averaging step that requires multiple passes through the model instead, and then are only approximate, which would introduce sampling bias here. In contrast, to sample from our method requires only a *single* pass through $\gamma_\theta$ and $k_\theta$, and involves no bias. The canonicalisation method of Kaba et al. (2023) also avoids this averaging step, but instead suffers from pathologies associated with its analogue of the equivariant base case $\gamma_\theta$, which Dym et al. (2024) show must become discontinuous at certain inputs. Additionally, canonicalisation requires an intrinsically $\mathcal{G}$-equivariant architecture for its analogue of $\gamma_\theta$. In contrast, whenever $\mathcal{G}$ is compact, our stochastic approach allows $\gamma_\theta$ to be obtained using the Haar measure in a way that does not suffer these pathologies, and may be implemented without any intrinsically $\mathcal{G}$-equivariant neural network components at all, as we show concretely next.

### 3.3 SYMDIFF FOR $N$-BODY SYSTEMS

We now apply SYMDIFF in the setting of $N$-body systems considered in Section 2.3. Specifically, we take $\mathcal{Z} \coloneqq \mathcal{U}$, $\mathcal{H} \coloneqq S_N$, and $\mathcal{G} \coloneqq \mathrm{O}(3)$, and consider the action on $\mathcal{Z}$ defined in equation 6. This means that we start with $k_\theta(\mathbf{z}_{t-1}|\mathbf{z}_t)$ in equation 8 that is already equivariant with respect to reorderings of the $N$ bodies, and then symmetrise this to obtain an $(S_N \times \mathrm{O}(3))$-equivariant reverse kernel overall. We choose to symmetrise in this way because highly scalable $S_N$-equivariant kernels based on Transformer architectures can be readily constructed for this purpose (Vaswani et al., 2017; Lee et al., 2019; Peebles & Xie, 2023), whereas intrinsically $\mathrm{O}(3)$-equivariant neural networks have not shown the same degree of scalability to-date (Abramson et al., 2024).

**Choice of unsymmetrised kernels** It remains now to choose $k_\theta(\mathbf{z}_{t-1}|\mathbf{z}_t)$. We now do so in a way that equation 9 will resemble the standard diffusion objective in Ho et al. (2020), allowing for the scalable training of SYMDIFF. Specifically, we take

$$k_\theta(\mathbf{z}_{t-1}|\mathbf{z}_t) \coloneqq \mathcal{N}_\mathcal{U}(\mathbf{z}_{t-1}; \mu_\theta(\mathbf{z}_t), \sigma_q^2(t)\mathbf{I}), \tag{10}$$

where $\mu_\theta : \mathcal{Z} \to \mathcal{Z}^3$ is an arbitrary $S_N$-equivariant neural network, which in turn means $k_\theta$ is stochastically $S_N$-equivariant. We highlight that $\mu_\theta$ is otherwise unconstrained and can process $[\mathbf{x}, \mathbf{h}]$ jointly. In contrast, previous work using intrinsically $(S_n \times \mathrm{O}(3))$-equivariant components (Satorras et al., 2021; Thölke & De Fabritiis, 2022; Hua et al., 2024) has required complex parameterisations for $\mu_\theta$ that handle the $\mathbf{x}$ and $\mathbf{h}$ inputs separately.

**Form of KL terms** We now show how our model yields a closed-form expression for the $\mathcal{L}_t$ terms in equation 9. Standard arguments show that each $q(\mathbf{z}_{t-1}|\mathbf{z}_t, \mathbf{z}_0) = \mathcal{N}_\mathcal{U}(\mathbf{z}_t; \mu_q(\mathbf{z}_t, \mathbf{z}_0), \sigma_q^2(t)\mathbf{I})$ is a (projected) Gaussian. We claim that our model also gives a (projected) Gaussian

$$k_\theta(\mathbf{z}_{t-1}|R, \mathbf{z}_t) = \mathcal{N}_\mathcal{U}(\mathbf{z}_{t-1}; R \cdot \mu_\theta(R^T \cdot \mathbf{z}_t), \sigma_q^2(t)\mathbf{I}). \tag{11}$$

Indeed, by the definition of this kernel in Proposition 1 and the definition of $\mathcal{N}_\mathcal{U}$ in Section 2.3, and since $R^{-1} = R^T$ for $R \in \mathrm{O}(3)$, for $\mathbf{z}_{t-1} \sim k_\theta(\mathbf{z}_{t-1}|R, \mathbf{z}_t)$ we have

$$\mathbf{z}_{t-1} \overset{\mathrm{d}}{=} R \cdot \left(\mu_\theta(R^T \cdot \mathbf{z}_t) + \sigma_q(t)\,\epsilon\right) = R \cdot \mu_\theta(R^T \cdot \mathbf{z}_t) + \sigma_q(t)R \cdot \epsilon,$$

where $\epsilon \sim \mathcal{N}_\mathcal{U}(0, \mathbf{I})$. Since $R \cdot \epsilon \overset{\mathrm{d}}{=} \epsilon$ for $R \in \mathrm{O}(3)$, equation 11 now follows. The same argument as Hoogeboom et al. (2022) now yields the closed-form expression

$$\mathcal{L}_t = \mathbb{E}_{q(\mathbf{z}_t|\mathbf{z}_0), \gamma_\theta(dR|\mathbf{z}_t)} \left[\frac{1}{2\sigma_q^2(t)} \left\|\mu_q(\mathbf{z}_t, \mathbf{z}_0) - R \cdot \mu_\theta(R^T \cdot \mathbf{z}_t)\right\|^2\right]. \tag{12}$$

---

[3]We leave the dependence on $t$ implicit in our notation throughout.

We can obtain unbiased gradients of this quantity whenever $\gamma_\theta(dR|\mathbf{z}_t)$ is *reparametrisable* (Kingma, 2013). In other words, we should define $\gamma_\theta(dR|\mathbf{z}_t)$ to be the distribution of $\varphi_\theta(\mathbf{z}_t, \xi)$, where $\varphi_\theta$ is a deterministic neural network, and $\xi$ is some noise variable whose distribution does not depend on $\theta$. For a discussion of how we can handle the $\mathcal{L}_1$ term in equation 9 in our framework, we refer to Appendix C.1.

**$\epsilon$-parameterisation**   When $\mu_\theta$ is taken to have the $\epsilon$-form (Ho et al., 2020; Kingma et al., 2021)

$$\mu_\theta(\mathbf{z}_t) := \frac{1}{\alpha_{t|t-1}}\mathbf{z}_t - \frac{\sigma^2_{t|t-1}}{\alpha_{t|t-1}\sigma_t}\epsilon_\theta(\mathbf{z}_t), \tag{13}$$

for some neural network $\epsilon_\theta : \mathcal{Z} \to \mathcal{Z}$, the same argument given by Ho et al. (2020) now allows us to rewrite equation 12 in a way that resembles the standard diffusion objective:

$$\mathcal{L}_t = \mathbb{E}_{q(\mathbf{z}_0), \epsilon \sim \mathcal{N}_\mathcal{U}(0,\mathbf{I}), \gamma_\theta(dR|\mathbf{z}_t)}\left[\frac{1}{2}w(t)\big\|\epsilon - R \cdot \epsilon_\theta(R^T \cdot \mathbf{z}_t)\big\|^2\right] \tag{14}$$

where $\mathbf{z}_t = \alpha_t\mathbf{z}_0 + \sigma_t\epsilon$, and $w(t) = (1 - \mathrm{SNR}(t-1)/\mathrm{SNR}(t))$. Recall we require $\mu_\theta$ to be $S_N$-equivariant, which straightforwardly follows here whenever $\epsilon_\theta$ is. In practice, we set $w(t) = 1$ during training as is commonly done in the diffusion literature (Kingma & Gao, 2024).

**Recursive choice of $\gamma_\theta$**   To apply Theorem 1, we require an equivariant base case $\gamma_\theta$ that is $S_N$-invariant and O(3)-equivariant. To obtain this, we apply the recursive procedure from equation 7, where $\gamma_0 : \mathcal{Z} \to \mathrm{O}(3)$ is obtained using the Haar measure on O(3) as described there, and $\gamma_1(dR|\mathbf{z}_t)$ is defined as the distribution of $f_\theta(\mathbf{z}_t, \eta)$, where $\eta$ is sampled from some noise distribution $\nu(d\eta)$ and $f_\theta(\cdot, \eta) : \mathcal{Z} \to \mathrm{O}(3)$ is a $S_N$-invariant neural network for each fixed value of $\eta$. Both the Haar measure and $\nu$ do not depend on $\theta$, and so the overall $\gamma_\theta$ obtained in this way is always reparametrisable by construction. We emphasise that $f_\theta$ is *not* required to be O(3)-equivariant in any sense, thus allowing for highly flexible choices such as Set Transformers (Lee et al., 2019). At sampling time, we use the procedure from Section 5 of Mezzadri (2006) to sample from the Haar measure on O(3), which is a negligible overhead compared with the cost of evaluating $f_\theta$.

---

**Algorithm 1** SYMDIFF training step

---

1: Sample $\mathbf{z}_0 \sim p_\mathrm{data}(\mathbf{z}_0)$, $t \sim \mathrm{Unif}(\{1, \ldots, T\})$ and $\epsilon \sim \mathcal{N}_\mathcal{U}(0, \mathbf{I})$
2: $\mathbf{z}_t \leftarrow \alpha_t\mathbf{z}_0 + \sigma_t\epsilon$
3: Sample $R_0$ from the Haar measure on O(3) and $\eta \sim \nu(d\eta)$
4: $R \leftarrow R_0 \cdot f_\theta(R_0^T \cdot \mathbf{z}_t, \eta)$
5: Take gradient descent step with $\nabla_\theta \frac{1}{2}w(t)\big\|\epsilon - R \cdot \epsilon_\theta(R^T \cdot \mathbf{z}_t)\big\|^2$

---

**Algorithm 2** SYMDIFF sampling process

---

1: Sample $\mathbf{z}_T \sim \mathcal{N}_\mathcal{U}(0, \mathbf{I})$
2: **for** $s = T, \ldots, 2$ **do**
3:     Sample $R_0$ from the Haar measure on O(3), $\eta \sim \nu(d\eta)$, and $\epsilon \sim \mathcal{N}_\mathcal{U}(0, \mathbf{I})$
4:     $R \leftarrow R_0 \cdot f_\theta(R_0^T \cdot \mathbf{z}_t, \eta)$
5:     $\mathbf{z}_{t-1} \leftarrow \frac{1}{\alpha_{t|t-1}}\mathbf{z}_t - \frac{\sigma^2_{t|t-1}}{\alpha_{t|t-1}\sigma_t}R \cdot \epsilon_\theta(R^T \cdot \mathbf{z}_t) + \sigma_q(t)\epsilon$
6: **return** $\mathbf{z}_0 \sim p_\theta(\mathbf{z}_0|\mathbf{z}_1)$                    $\triangleright$ See Appendix C.1 for an example of this output kernel

---

### 3.4   DATA AUGMENTATION IS A SPECIAL CASE OF SYMDIFF

Data augmentation is a popular method for incorporating "soft" inductive biases within neural networks. Consider again our setup from the previous section, but now using the unsymmetrised kernels $k_\theta(\mathbf{z}_{t-1}|\mathbf{z}_t)$ from equation 10 in place of $p_\theta(\mathbf{z}_{t-1}|\mathbf{z}_t)$ in our backwards process. Suppose this model is trained using the standard diffusion objective, applying a uniform random orthogonal transformation to the input of $\epsilon_\theta$ before each forward pass. This is equivalent to optimising the

following objective:

$$\mathcal{L}_t^{\text{aug}} = \mathbb{E}_{q(\mathbf{z}_0), \epsilon \sim \mathcal{N}_{\mathcal{U}}(0, \mathbf{I}), \lambda(dR)} \left[ \frac{1}{2} w(t) \|\epsilon - \epsilon_\theta (\alpha_t R \cdot \mathbf{z}_0 + \sigma_t \epsilon)\|^2 \right], \qquad (15)$$

where $\lambda$ is the Haar measure on O(3). (More general choices of $\lambda$ could also be considered.) We then have the following result, proven in Appendix A.3.

**Proposition 2.** *When $\gamma_\theta(dR|\mathbf{z}_t) = \lambda(dR)$ for all $\mathbf{z}_t \in \mathcal{Z}$, our* SYMDIFF *objective in equation 14 recovers the data augmentation objective exactly, so that $\mathcal{L}_t = \mathcal{L}_t^{\text{aug}}$.*

As a result, SYMDIFF may be understood as a strict generalisation of data augmentation in which a more flexible augmentation process $\gamma_\theta$ is *learned* during training. Moreover, our $\gamma_\theta$ is then also deployed at sampling time in a way that guarantees stochastic equivariance. In contrast, data augmentation is usually only applied during training, and the model deployed at sampling time then becomes only approximately equivariant.

## 4 EXPERIMENTS

We evaluated our $E(3)$-equivariant SYMDIFF model as a drop-in replacement for the $E(3)$-equivariant diffusion (EDM) of Hoogeboom et al. (2022) on both the QM9 and GEOM-Drugs datasets for molecular generation. We implemented this within the official codebase of Hoogeboom et al. (2022)[4] substituting our symmetrised reverse process for their one. We made minimal other changes to their code and experimental setup otherwise, and performed minimal tuning of our architecture. We found SYMDIFF led to significantly improved performance on both tasks. Our results were also on par or better than GeoLDM (Xu et al., 2023), END (Cornet et al., 2024), and MUDiff (Hua et al., 2024), which bake in more sophisticated inductive biases related to molecular generation than EDM does. Our models were also more computationally efficient than these baselines, which all rely on intrinsically equivariant subcomponents. We give an overview of our setup now and provide full details in Appendix C. Our code is available at: https://github.com/leozhangML/SymDiff.

### 4.1 QM9

**Dataset** QM9 (Ramakrishnan et al., 2014) is a common benchmark dataset used for evaluating molecular generation. It consists of molecular properties and atom coordinates for 130k small molecules with up to 9 heavy atoms and a total of 29 atoms including hydrogen. For our experiments, we trained our SYMDIFF method to generate molecules with 3D coordinates, atom types, and atom charges where we explicitly modeled hydrogen atoms. We used the same train-val-test split of 100K-8K-13K as in Anderson et al. (2019).

**Our model** We took our core model (SymDiff) to be $\text{sym}_{\gamma_\theta}(k_\theta)(\mathbf{z}_{t-1}|\mathbf{z}_t)$ from equation 8 above, with $k_\theta$ and $\gamma_\theta$ given as specified in Section 3.3. We chose the "backbone" neural network $\epsilon_\theta$ to be a Diffusion Transformer (DiT) (Peebles & Xie, 2023), which is $S_N$-equivariant by construction. Likewise, we chose the "backbone" neural network $f_\theta$ of the component $\gamma_\theta$ to be a DiT. We made this component $S_n$-invariant using a Set Transformer (Lee et al., 2019) approach, thereby achieving the requirements described in Section 3.3. Our $\epsilon_\theta$ had 29M parameters, matching the smallest model considered by Peebles & Xie (2023), while our $f_\theta$ had 2.2M parameters. In this way, our $\gamma_\theta$ was much smaller than our $k_\theta$, following a similar approach taken by earlier work on deterministic symmetrisation (Kim et al., 2023; Kaba et al., 2023). Overall, our model had 31.2M parameters in total. To test its scalability, we also trained a larger version of our method with a backbone $k_\theta$ having 115.6M parameters (SymDiff*), which matched the DiT-B model from Peebles & Xie (2023). We trained all our models for 4350 epochs to match the same number of gradient steps as Hoogeboom et al. (2022). For further details about our architecture, see Appendix B.

**Metrics** To measure the quality of generated molecules, we follow standard practice (Hoogeboom et al., 2022; Garcia Satorras et al., 2021a) and report atom stability, molecular stability, validity

---

[4]https://github.com/ehoogeboom/e3_diffusion_for_molecules

and uniqueness. We exclude results for the novelty metric for the same reasons as discussed in Vignac & Frossard (2021) and refer the reader to these works for a more extensive discussion of these metrics. For all of our metrics, we used 10,000 samples and report the mean and standard deviation over three evaluation runs. To demonstrate the efficiency of our approach, we also report the number of seconds per epoch, time taken to generate one sample and vRAM.

**Baselines** As a baseline, we trained the 5.3M parameter EDM model using the original experimental setup of Hoogeboom et al. (2022). We also trained an unsymmetrised reverse process with a 29M parameter DiT backbone (DiT), as well as the same model using data augmentation as described in Section 3.4 (DiT-Aug). Additionally, we trained a SYMDIFF model with the same backbone $k_\theta$ as our SymDiff model above, but whose $\gamma_\theta$ was obtained using the Haar measure on $O(3)$, rather than learning this component (SymDiff-H).

**Results** From Table 1, we see that our SYMDIFF models comfortably outperformed EDM on all metrics, bar uniqueness. Additionally, our model was also competitive with the more recent, sophisticated baselines from the literature, outperforming all of them on validity. We attribute the improved performance of our method to the extra architectural flexibility provided by our approach to symmetrisation. Our largest model, SymDiff*, outperformed all our baselines on atom stability and validity, and is within variance for molecular stability. We conjecture that similar performance improvements could be achieved by using our SYMDIFF approach as a drop-in replacement for the reverse kernels in more sophisticated methods. Table 1 also shows that DiT-Aug performed notably better than the DiT model on all metrics, highlighting its strength as a baseline. Despite this, our SymDiff model outperformed both SymDiff-H and DiT-Aug on all metrics apart from uniqueness. This shows that our approach has benefits that extend beyond merely performing data augmentation.

**Computational efficiency** Importantly, as Table 2 shows, our method was also more computationally efficient than the alternative methods we considered, both in terms of seconds/epoch, sampling time, and vRAM. This is not surprising since these alternative models rely on intrinsically equivariant graph neural networks that use message passing during training and inference, which is computationally very costly. In contrast, our symmetrisation approach allows us to use computationally efficient DiT components that parallelise and scale much more effectively (Fei et al., 2024).

Table 1: Test NLL, atom stability, molecular stability, validity and uniqueness on QM9 for 10,000 samples and 3 evaluation runs. We omit the results for NLL where not available.

| Method | NLL ↓ | Atm. stability (%) ↑ | Mol. stability (%) ↑ | Val. (%) ↑ | Uniq. (%) ↑ |
|---|---|---|---|---|---|
| GeoLDM | – | $98.90_{\pm0.10}$ | $89.40_{\pm0.50}$ | $93.80_{\pm0.40}$ | $92.70_{\pm0.50}$ |
| MUDiff | $\mathbf{-135.50}_{\pm2.10}$ | $98.80_{\pm0.20}$ | $\mathbf{89.90}_{\pm1.10}$ | $95.30_{\pm1.50}$ | $\mathbf{99.10}_{\pm0.50}$ |
| END | – | $98.90_{\pm0.00}$ | $89.10_{\pm0.10}$ | $94.80_{\pm0.10}$ | $92.60_{\pm0.20}$ |
| EDM | $-110.70_{\pm1.50}$ | $98.70_{\pm0.10}$ | $82.00_{\pm0.40}$ | $91.90_{\pm0.50}$ | $90.70_{\pm0.60}$ |
| SymDiff* | $\mathbf{-133.79}_{\pm1.33}$ | $\mathbf{98.92}_{\pm0.03}$ | $\mathbf{89.65}_{\pm0.10}$ | $\mathbf{96.36}_{\pm0.27}$ | $97.66_{\pm0.22}$ |
| SymDiff | $-129.35_{\pm1.07}$ | $98.74_{\pm0.03}$ | $87.49_{\pm0.23}$ | $95.75_{\pm0.10}$ | $97.89_{\pm0.26}$ |
| SymDiff-H | $-126.53_{\pm0.90}$ | $98.57_{\pm0.07}$ | $85.51_{\pm0.18}$ | $95.22_{\pm0.18}$ | $97.98_{\pm0.09}$ |
| DiT-Aug | $-126.81_{\pm1.69}$ | $98.64_{\pm0.03}$ | $85.85_{\pm0.24}$ | $95.10_{\pm0.17}$ | $97.98_{\pm0.08}$ |
| DiT | $-127.78_{\pm2.49}$ | $98.23_{\pm0.04}$ | $81.03_{\pm0.25}$ | $94.71_{\pm0.31}$ | $97.98_{\pm0.12}$ |
| Data | | 99.00 | 95.20 | 97.8 | 100 |

Table 2: Seconds per epoch, sampling time and vRAM for SymDiff and our baselines on QM9. Results for END are omitted as their code was not publicly available.

| Method | # Parameters | Sec./epoch (s) ↓ | Sampling time (s) ↓ | vRAM (GB) ↓ |
|---|---|---|---|---|
| GeoLDM | 11.4M | 210.93 | 0.26 | 27 |
| MuDiff | 9.7M | 230.87 | 0.89 | 36 |
| END | 9.4M | – | – | – |
| EDM | 5.4M | 88.80 | 0.27 | 14 |
| SymDiff* | 117.8M | 53.40 | 0.21 | 16 |
| SymDiff | 31.2M | 27.20 | 0.09 | 7 |

**Ablations** As an ablation study, we also tested the effect of making SYMDIFF smaller, and EDM larger. For SymDiff, we trained two models of 23.5M (SymDiff$^-$) and 13.5M (SymDiff$^{--}$)

parameters respectively. For EDM, we trained two additional models with 9.5M (EDM$^+$) and 12.4M (EDM$^{++}$) parameters respectively. For full details see Appendix C.2.1. From Table 3, we see that even our smaller SymDiff models remained competitive. In particular, SymDiff$^-$ gave comparable molecular stability as the second largest EDM model, EDM$^+$, while being approximately 5 times faster in terms of seconds/epoch.

Table 3: NLL, molecular stability, seconds per epoch, sampling time and vRAM for different sizes of SymDiff and EDM on QM9. For additional performance metrics see Appendix C.2.

| Method | NLL $\downarrow$ | Mol. stability (%) $\uparrow$ | Sec./epoch (s) $\downarrow$ | Sampling time (s) $\downarrow$ | vRAM (GB) $\downarrow$ |
|---|---|---|---|---|---|
| EDM$^{++}$ | -119.12$_{\pm1.41}$ | 85.68$_{\pm0.83}$ | 160.60 | 0.56 | 23 |
| EDM$^+$ | -110.97$_{\pm1.42}$ | 84.63$_{\pm0.16}$ | 192.60 | 0.46 | 23 |
| EDM | -110.70$_{\pm1.50}$ | 82.00 $_{\pm0.40}$ | 88.80 | 0.27 | 14 |
| SymDiff | -129.35$_{\pm1.07}$ | 87.49$_{\pm0.23}$ | 27.20 | 0.09 | 7 |
| SymDiff$^-$ | -125.40$_{\pm0.63}$ | 83.51$_{\pm0.24}$ | 24.87 | 0.08 | 6 |
| SymDiff$^{--}$ | -110.68 $_{\pm2.55}$ | 71.25 $_{\pm0.50}$ | 20.60 | 0.07 | 5 |

## 4.2 GEOM-DRUGS

**Dataset, model and training**   GEOM-Drugs (Axelrod & Gomez-Bombarelli, 2022) is a larger and more complicated dataset than QM9, containing 430,000 molecules with up to 181 atoms. We processed the dataset in the same way as Hoogeboom et al. (2022), where we again model hydrogen explicitly. We used the SymDiff model from earlier, which we trained for 55 epochs to match the same number of gradient steps as Hoogeboom et al. (2022).

**Metrics and baselines**   We report the same metrics as for QM9 but exclude molecular stability and uniqueness for the same reasons discussed in Hoogeboom et al. (2022). We compared our method to the EDM model used by Hoogeboom et al. (2022) for GEOM-Drugs, as well as the other baseline architectures reported for QM9.

**Results**   From Table 4 we again see that our approach comfortably outperformed its EDM counterpart. It is also again competitive with the more sophisticated baselines, whose reported results we restate here. Like with QM9, our SymDiff models were significantly less costly in terms of compute time and memory usage compared with EDM (see Appendix C.3). In fact, when we tried to run the EDM model it resulted in out-of-memory errors on our NVIDIA H100 80GB GPU (Hoogeboom et al. (2022) avoid this by training EDM on $3\times$ NVIDIA RTX A6000 48GB GPUs.)

Table 4: Test NLL, atom stability and validity on GEOM-Drugs for 10,000 samples and 3 evaluation runs. GeoLDM and EDM ran their results for just one evaluation run. We omit the results for NLL and validity where not available.

| Method | NLL $\downarrow$ | Atm. stability (%) $\uparrow$ | Val. (%) $\uparrow$ |
|---|---|---|---|
| GeoLDM | – | 84.4 | 99.3 |
| END | – | **87.8**$_{\pm0.99}$ | 92.9$_{\pm0.3}$ |
| EDM | -137.1 | 81.3 | – |
| SymDiff | **-301.21**$_{\pm0.53}$ | 86.16$_{\pm0.05}$ | 99.27$_{\pm0.1}$ |
| Data | | 86.50 | 99.9 |

## 5 CONCLUSION

We have introduced SYMDIFF: a lightweight, and scalable framework for constructing equivariant diffusion models based on stochastic symmetrisation. We applied this approach to E(3)-equivariance for $N$-body data, obtaining an overall model that is stochastically equivariant but that does not rely on any intrinsically equivariant neural network subcomponents. Our approach leads to significantly greater modelling flexibility, which allows leveraging powerful off-the-shelf architectures such as Transformers (Vaswani et al., 2017). We showed empirically that this leads overall to improved performance on several relevant benchmarks.

ACKNOWLEDGMENTS

The authors are grateful to Tom Rainforth, Emile Mathieu, Saifuddin Syed and Ahmed Elhag for helpful discussions. LZ and KA are supported by the EPSRC CDT in Modern Statistics and Statistical Machine Learning (EP/S023151/1).

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

# A PROOFS

## A.1 PROOF OF THEOREM 1

*Proof.* Theorem 1 is a special case of Example 6.3 of Cornish (2024), whose notation and setup we will import freely here. Note that Example 6.3 makes use of *string diagrams* (Selinger, 2010), an introduction to which can be found in Section 2 of Cornish (2024). Intuitively, a string diagram represents a (possibly stochastic) computational processes that should be read up the page, with the inputs applied at the bottom, and outputs produced at the top.

To proceed, we first let the action $\rho$ be trivial, so that the semidirect product $N \rtimes_\rho H$ becomes simply the direct product $N \times H$ by Remark 3.29 of Cornish (2024). Now consider the following string diagram:

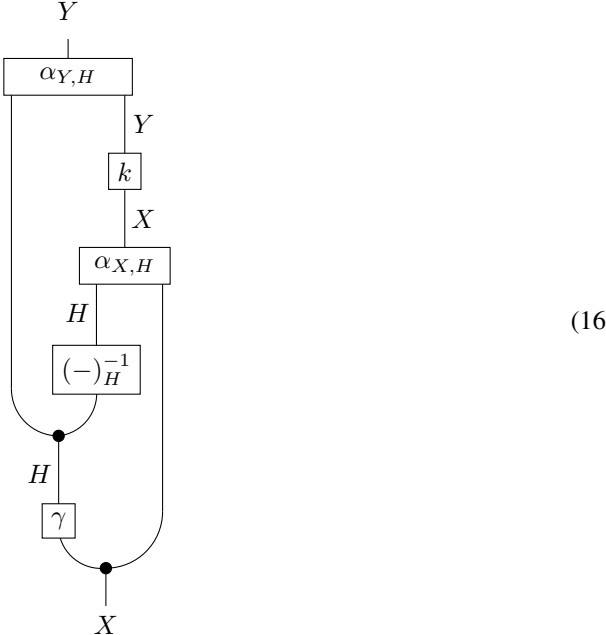

$$(16)$$

The upshot of Example 6.3 of Cornish (2024) is that equation 16 is always equivariant with respect to the $(N \times H)$-actions $\alpha_X$ and $\alpha_Y$ whenever the following conditions all hold:

- $\gamma : X \to H$ is equivariant to the $H$-actions $\alpha_{X,H}$ and $*_H$;

- $\gamma : X \to H$ is invariant to the $N$-action $\alpha_{X,N}$;

- $k : X \to Y$ is equivariant with respect to the $N$-actions $\alpha_{X,N}$ and $\alpha_{Y,N}$.

Here, as in Example 6.3 of Cornish (2024):

- We have decomposed the $(N \times H)$-action $\alpha_X$ as an $H$-action $\alpha_{X,H}$ followed by an $N$-action $\alpha_{X,N}$ using Remark 3.31 of Cornish (2024). In other words:

- We have decomposed $\alpha_Y$ similarly;

- We denote by $*_H$ and $(-)_H^{-1}$ the multiplication and inversion operations of the group $H$.

To obtain our Theorem 1, we now simply instantiate things appropriately in the Markov category $\mathsf{C} := \mathsf{Stoch}$, so that all the boxes appearing in equation 16 become Markov kernels. Concretely, we take $X := \mathcal{X}$, $Y := \mathcal{Y}$, $N := \mathcal{H}$, and $H := \mathcal{G}$. A procedure for sampling from the symmetrised Markov kernel in equation 16 may then be read off as follows:

$$g \sim \gamma(dg|\mathbf{x}) \qquad \mathbf{y} \sim k(d\mathbf{y}|g^{-1} \cdot \mathbf{x}) \qquad \text{return } g \cdot \mathbf{y},$$

exactly as in the statement of this result. □

## A.2 PROOF OF PROPOSITION 1

*Proof.* For simplicity, we assume $k(d\mathbf{y}|\mathbf{x})$ has a density $k(\mathbf{y}|\mathbf{x})$ with respect to the Lebesgue measure $\mu$ on $\mathcal{Y} = \mathbb{R}^n$. To derive the density of $k(d\mathbf{y}|g, \mathbf{x})$ where $g \in \mathcal{G}$ is some fixed group element, we note that for $\mathbf{y} \sim k(d\mathbf{y}|g, \mathbf{x})$, we have $\mathbf{y} = g \cdot \mathbf{y}_0$ where $\mathbf{y}_0 \sim k(\mathbf{y}|g^{-1} \cdot \mathbf{x})$. Hence, by the change-of-variables formula, we can conclude that the density of $k(\mathbf{y}|g, \mathbf{x})$ exists and has the form:

$$k(\mathbf{y}|g, \mathbf{x}) = k(g^{-1} \cdot \mathbf{y}|g^{-1} \cdot \mathbf{x}) \left| \frac{\partial(g^{-1} \cdot \mathbf{y})}{\partial \mathbf{y}} \right|.$$

Hence, we see that when the action of $\mathcal{G}$ has unit Jacobian, the density of $k(\mathbf{y}|g, \mathbf{x}) = k(g^{-1} \cdot \mathbf{y}|g^{-1} \cdot \mathbf{x})$.

Further, suppose we have $g \sim \gamma(dg|\mathbf{x})$ and $\mathbf{y} \sim k(\mathbf{y}|g, \mathbf{x})$ - i.e. $\mathbf{y} \sim \mathsf{sym}_\gamma(k)(d\mathbf{y}|\mathbf{x})$. It is the case that for an arbitrary (Borel) measurable set $A$ in $\mathcal{Y}$, we have

$$\mathsf{sym}_\gamma(k)(\mathbf{y} \in A|\mathbf{x}) = \int_\mathcal{G} k(\mathbf{y} \in A|g, \mathbf{x}) \, \gamma(dg|\mathbf{x}).$$

Since we have shown above that $k(d\mathbf{y}|g, \mathbf{x})$ has a density, we can express this as

$$\mathsf{sym}_\gamma(k)(\mathbf{y} \in A|\mathbf{x}) = \int_\mathcal{G} \left( \int_A k(\mathbf{y}|g, \mathbf{x}) \, \mu(d\mathbf{y}) \right) \gamma(dg|\mathbf{x})$$

$$= \int_A \mathbb{E}_{\gamma(dg|\mathbf{x})} \left[ k(\mathbf{y}|g, \mathbf{x}) \right] \mu(d\mathbf{y}).$$

where we use Fubini's theorem for the second line as all quantities are non-negative. Hence, we can conclude that the density of $\mathsf{sym}_\gamma(k)$ exists and has the form $\mathsf{sym}_\gamma(k) = \mathbb{E}_{\gamma(dg|\mathbf{x})}[k(\mathbf{y}|g, \mathbf{x})]$. □

## A.3 PROOF OF PROPOSITION 2

*Proof.* The standard diffusion objective with data augmentation distributed according to the Haar measure $\lambda$ is given by

$$\mathcal{L}_t^{\text{aug}} = \mathbb{E}_{q(\mathbf{z}_0), \epsilon \sim \mathcal{N}_\mathcal{U}(0, \mathbf{I}), \lambda(dR)} \left[ \frac{1}{2} w(t) \| \epsilon - \epsilon_\theta(\alpha_t R \cdot \mathbf{z}_0 + \sigma_t \epsilon) \|^2 \right] \tag{17}$$

$$= \mathbb{E}_{q(\mathbf{z}_0), \epsilon \sim \mathcal{N}_\mathcal{U}(0, \mathbf{I}), \lambda(dR)} \left[ \frac{1}{2} w(t) \| \epsilon - \epsilon_\theta(R \cdot (\alpha_t \mathbf{z}_0 + \sigma_t R^T \cdot \epsilon)) \|^2 \right] \tag{18}$$

$$= \mathbb{E}_{q(\mathbf{z}_0), \epsilon' \sim \mathcal{N}_\mathcal{U}(0, \mathbf{I}), \lambda(dR)} \left[ \frac{1}{2} w(t) \| R \cdot \epsilon' - \epsilon_\theta(R \cdot (\alpha_t \mathbf{z}_0 + \sigma_t \epsilon')) \|^2 \right] \tag{19}$$

$$= \mathbb{E}_{q(\mathbf{z}_0), \epsilon \sim \mathcal{N}_\mathcal{U}(0, \mathbf{I}), \lambda(dR)} \left[ \frac{1}{2} w(t) \| \epsilon - R^T \cdot \epsilon_\theta(R \cdot (\alpha_t \mathbf{z}_0 + \sigma_t \epsilon)) \|^2 \right], \tag{20}$$

where we use the fact that $\epsilon' = R^T \cdot \epsilon$ is distributed according to $\mathcal{N}_\mathcal{U}(0, \mathbf{I})$ as $R, \epsilon$ are independent in the expectation, and that the action of $R \in \mathrm{O}(3)$ preserves the L2 norm. To conclude, we note that it is a standard result that if $R \sim \lambda$, the inverse $R^T$ is also distributed according to the Haar measure $\lambda$. Hence, we see that $\mathcal{L}_t^{\text{aug}}$ coincides with

$$\mathcal{L}_t = \mathbb{E}_{q(\mathbf{z}_0), \epsilon \sim \mathcal{N}_\mathcal{U}(0, \mathbf{I}), \lambda(dR)} \left[ \frac{1}{2} w(t) \| \epsilon - R \cdot \epsilon_\theta(R^T \cdot \mathbf{z}_t) \|^2 \right], \tag{21}$$

where $\mathbf{z}_t = \alpha_t \mathbf{z}_0 + \sigma_t \epsilon$. □

# B MODEL ARCHITECTURE

Below, we outline the architectures used for $\epsilon_\theta$ and $\gamma_\theta$. Both components rely on Diffusion Transformers (DiTs) (Peebles & Xie, 2023) using the official PyTorch implementation at `https://github.com/facebookresearch/DiT`. We also state the hyperparameters that we kept fixed for both our QM9 and GEOM-Drugs experiments. Any hyperparameters that differed between the datasets are discussed in their respective sections later in the Appendix.

We emphasise that our architecture choices were not extensively tuned as the main purpose of our experiments was to show that we can use generic architectures for equivariant diffusion models. We arrived at the below architecture through small adjustments from experimenting with DiT models in the context of molecular generation, which we stuck with for our final experiments.

## B.1 ARCHITECTURE OF $\epsilon_\theta$

As we need $\epsilon_\theta$ to be an $S_N$-equivariant architecture, we parametrise this in terms of a DiT model which consists of $n_{\text{layers}}$ (intermediate) layers, $n_{\text{head}}$ attention heads, hidden size $n_{\text{size}}$ and a final output layer. We then project the outputs via $\text{proj}_{\mathcal{U}}$ to ensure the outputs lie in $\mathcal{U}$. In addition, for the MLP layers, we use SwiGLU activations (Shazeer, 2020) instead of the standard GELU, where the ratio of the hidden size of the SwiGLU to $n_{\text{size}}$ is 2, and we do not use the default Fourier embeddings for the inputs - we pass our inputs directly into the model. We also use the default time embeddings. We refer to this model setup as `DiT`.

We also use the Gaussian positional embeddings from Luo et al. (2022) as additional features that we concatenate to the inputs of `DiT`. To compute this from $\mathbf{x}$, we let

$$\psi_{(i,j)}^k = -\frac{1}{\sqrt{2\pi}|\sigma^k|} \exp\left(-\frac{1}{2}\left(\frac{\left\|\mathbf{x}^{(j)} - \mathbf{x}^{(j)}\right\| - \mu^k}{|\sigma^k|}\right)\right),$$

where $k = 1, \ldots, K$ is the number of basis kernels we use and $\mu^k, \sigma^k \in \mathbb{R}$ are learnable parameters. We define $\psi_{(i,j)} = (\psi_{(i,j)}^1, \ldots, \psi_{(i,j)}^K)^T \in \mathbb{R}^{K \times 1}$. We then compute our positional embeddings by $\Psi_i = \frac{1}{N} \sum_{j=1}^N \psi_{(i,j)} W_D$ where $W_D \in \mathbb{R}^{K \times n_{\text{emb}}}$ is a learnable matrix, and we concatenate these to form our embeddings $\boldsymbol{\Psi} = [\Psi_1, \ldots, \Psi_N] \in \mathbb{R}^{N \times n_{\text{emb}}}$. We note that $\boldsymbol{\Psi}$ is $O(3)$-invariant.

Finally, we provide pseudo-code for a single pass through $\epsilon_\theta$ in Algorithm 3 where we note that $W_I$ is a learnable linear layer.

---

**Algorithm 3** Computation of $\epsilon_\theta$

---

**Inputs:** $\mathbf{z} = [\mathbf{x}, \mathbf{h}]$ where $\mathbf{x} \in \mathbb{R}^{N \times 3}, \mathbf{h} \in \mathbb{R}^{N \times d}$; $t \in \mathbb{R}$
1: Compute $\boldsymbol{\Psi} \in \mathbb{R}^{N \times n_{\text{emb}}}$ from $\mathbf{x}$
2: $\mathbf{z} \leftarrow [\mathbf{x}, \mathbf{h}]W_I$ where $W_I \in \mathbb{R}^{(3+d) \times n_z}$
3: $\mathbf{z} \leftarrow \text{DiT}(t, [\mathbf{z}, \boldsymbol{\Psi}])$
4: **return** $\mathbf{z}$

---

## B.2 ARCHITECTURE OF $\gamma_\theta$

We construct $\gamma_\theta$ following the recursive setup in Section 3.3 where we take the noise distribution $\nu$ to be $\mathcal{N}_{\mathcal{U}}(0, \mathbf{I})$ on $\mathbb{R}^{N \times m_{\text{noise}}}$. We provide pseudo-code for a single pass through our $f_\theta$ in Algorithm 4 where we note that $W_G, W_1, W_2$ are learnable linear layers, we use the same embedding parameters for $\boldsymbol{\Psi}$ as before and `DiTWithoutFinalLayer` is the same as a `DiT` with $m_{\text{layers}}$ (intermediate) layers, $m_{\text{head}}$ attention heads, hidden size $m_{\text{size}}$ but where we do not apply the final layer.

## B.3 HYPERPARAMETERS FOR $\gamma_\theta$ AND $\epsilon_\theta$

For both QM9 and GEOM-Drugs, we fixed the following hyperparameters. For $\epsilon_\theta$, we set $K \approx \frac{1}{2}n_{\text{size}}$ and $n_{\text{emb}} = n_{\text{size}} - n_z$. For $\gamma_\theta$, we set $m_{\text{noise}} = 3$ and $m'_{\text{size}} = \frac{1}{2}m_{\text{size}}$.

---

**Algorithm 4** Computation of $f_\theta$

---

**Inputs:** $\mathbf{z} = [\mathbf{x}, \mathbf{h}]$ where $\mathbf{x} \in \mathbb{R}^{N \times 3}, \mathbf{h} \in \mathbb{R}^{N \times d}; \eta \in \mathbb{R}^{N \times m_{\text{noise}}}; t \in \mathbb{R}$

1: Compute $\mathbf{\Psi} \in \mathbb{R}^{N \times n_{\text{emb}}}$ from $\mathbf{x}$
2: $\mathbf{z} \leftarrow [\mathbf{x}, \eta, \mathbf{\Psi}]W_G$ where $W_G \in \mathbb{R}^{(3 + m_{\text{noise}} + n_{\text{emb}}) \times m_{\text{size}}}$
3: $\mathbf{z} \leftarrow \text{DiTWithoutFinalLayer}(t, \mathbf{z})$
4: $\mathbf{z} \leftarrow \frac{\mathbf{1}^\top \mathbf{z}}{N}$ where $\mathbf{1} \in \mathbb{R}^N$ is a vector of ones            $\triangleright$ Ensures $S_N$-invariance
5: $\mathbf{z} \leftarrow \text{GELU}(\mathbf{z}W_1)W_2$ where $W_1 \in \mathbb{R}^{m_{\text{size}} \times m'_{\text{size}}}, W_2 \in \mathbb{R}^{m'_{\text{size}} \times (3 \times 3)}$
6: $R \leftarrow \text{QRDecomposition}(\mathbf{z})$
7: **return** $R$

---

# C   EXPERIMENTAL DETAILS

## C.1   FIRST LIKELIHOOD TERM $\mathcal{L}_1$

We have a presented a framework for parametrising and optimising $p_\theta(\mathbf{z}_{t-1}|\mathbf{z}_t)$ for $t > 1$ in Section 3.3 where $p_\theta$ is obtained via stochastic symmetrisation. This corresponds to the $\mathcal{L}_t$ terms where $t > 1$ from our objective in equation 9. However, we note that standard diffusion models usually choose a different parametrisation for $p_\theta(\mathbf{z}_0|\mathbf{z}_1)$ as this corresponds to the final generation step. Depending on the modelling task, this requires a different approach compared to the other reverse kernels.

For example, in Hoogeboom et al. (2022), $p_\theta(\mathbf{z}_0|\mathbf{z}_1)$ is defined as the product of densities $p_\theta^{\text{cont}}(\mathbf{x}_0|\mathbf{z}_1)p_\theta^{\text{disc}}(\mathbf{h}_0|\mathbf{z}_1)$ where $p_\theta^{\text{disc}}$ implements a quantisation step converting the continuous latent $\mathbf{z}_1$ to discrete values $\mathbf{h}_0$, while $p_\theta^{\text{cont}}$ is still a Gaussian distribution generating continuous geometric features $\mathbf{x}_0$ from $\mathbf{z}_1$. In particular, we have that

$$p_\theta^{\text{cont}}(\mathbf{x}_0|\mathbf{z}_1) = \mathcal{N}_{\mathcal{U}}(\mathbf{x}_0; \mathbf{x}_1/\alpha_1 - \sigma_1/\alpha_1 \epsilon_\theta^{(\mathbf{x})}(\mathbf{z}_1), \sigma_1^2/\alpha_1^2 \mathbf{I})$$

where $\epsilon_\theta : \mathcal{Z} \to \mathcal{Z}$ is some $S_N$-invariant neural network and $\epsilon_\theta^{(\mathbf{x})}$ denotes the $\mathbf{x}$ component of the output of $\epsilon_\theta$.

We note that our proposed methodology can still account for this case by defining the symmetrised kernel by

$$p_\theta(\mathbf{z}_0|\mathbf{z}_1) = \text{sym}_{\gamma_\theta}(k_\theta)(\mathbf{z}_0|\mathbf{z}_1), \qquad k_\theta(\mathbf{z}_0|\mathbf{z}_1) = p_\theta^{\text{cont}}(\mathbf{x}_0|\mathbf{z}_1)p_\theta^{\text{disc}}(\mathbf{h}_0|\mathbf{z}_1)$$

We can follow the same discussion in Section 3.3 to conclude that

$$p_\theta(\mathbf{z}_0|\mathbf{z}_1) = \mathbb{E}_{\gamma_\theta(dR|\mathbf{z}_1)}[k_\theta(\mathbf{z}_0|R, \mathbf{z}_1)], \qquad k_\theta(\mathbf{z}_0|R, \mathbf{z}_1) = k_\theta^{\text{cont}}(\mathbf{x}_0|R, \mathbf{z}_1)k_\theta^{\text{disc}}(\mathbf{h}_0|R, \mathbf{z}_1),$$

where $k_\theta^{\text{cont}}(\mathbf{z}_0|R, \mathbf{z}_1) = \mathcal{N}(\mathbf{x}_0; \mathbf{x}_1/\alpha_1 - \sigma_1/\alpha_1 R \cdot \epsilon_\theta(R^T \cdot \mathbf{z}_1), \sigma_1^2/\alpha_1^2 \mathbf{I})$ and $k_\theta^{\text{disc}}(\mathbf{h}_0|R, \mathbf{z}_1) = p_\theta^{\text{disc}}(\mathbf{h}_0|R^T \cdot \mathbf{z}_1)$. This allows us to decompose $\mathcal{L}_1$ into the form in equation 9 and to tractably optimise this objective since we have access to the density $k_\theta(\mathbf{z}_0|R, \mathbf{z}_1)$.

## C.2   QM9 DETAILS

### C.2.1   MODEL HYPERPARAMETERS

For all of our experiments, we retain the diffusion hyperparameters as EDM (Hoogeboom et al., 2022) - i.e. we use the same noise schedule, discretisation steps etc.

**SymDiff**   Table 5 shows the hyperparameters for the $\epsilon_\theta$ backbone of the $k_\theta$ component used in the SYMDIFF models for QM9. The remaining hyperparameters were kept the same as in Appendix B.3.

For the $f_\theta$ backbone of the $\gamma_\theta$ component, we set $m_{\text{size}} = 128, m_{\text{layers}} = 8, m_{\text{heads}} = 4$ for all models bar SymDiff*. For SymDiff*, we set $m_{\text{size}} = 216, m_{\text{layers}} = 10, m_{\text{heads}} = 8$.

**EDM**   Table 6 shows the hyperparameters for the EDM models that we used for our QM9 experiments. The remaining model hyperparameters were kept the same as those in Hoogeboom et al. (2022).

Table 5: Choice of $n_{\text{size}}$, $n_{\text{layers}}$, $n_{\text{heads}}$ for the $\epsilon_\theta$ of the SYMDIFF models used for QM9.

| Model | # Parameters | $n_{\text{size}}$ | $n_{\text{layers}}$ | $n_{\text{heads}}$ |
|---|---|---|---|---|
| SymDiff* | 115.6M | 768 | 12 | 12 |
| SymDiff | 29M | 384 | 12 | 6 |
| SymDiff$^-$ | 21.3M | 360 | 10 | 6 |
| SymDiff$^{--}$ | 11.3M | 294 | 8 | 6 |

Table 6: Choices of the hyperparameters $n_f$ (# features per layer), $n_l$ (number of layers) for the EDM models used for QM9.

| Model | # Parameters | $n_f$ | $n_l$ |
|---|---|---|---|
| EDM$^{++}$ | 12.4M | 332 | 12 |
| EDM$^+$ | 9.5M | 256 | 16 |
| EDM | 5.3M | 256 | 9 |

### C.2.2 OPTIMISATION

For the optimisation of SYMDIFF models, we followed Peebles & Xie (2023) and used AdamW (Loshchilov & Hutter) with a batch size of 256. We chose a learning rate of $2 \times 10^{-4}$ and weight decay of $10^{-12}$ for our 31.2M parameter model by searching over a small grid of 3 values for each. To match the same number of steps as in Hoogeboom et al. (2022), we trained our model for 4350 epochs.

We applied the same optimization hyperparameters from our 31.2M model to all other SYMDIFF models bar SymDiff*, where we used a learning rate of $10^{-4}$. For the EDM models, we followed the default hyperparameters from Hoogeboom et al. (2022). In our augmentation experiments, we first tuned the learning rate and weight decay for the DiT model, keeping all other optimization hyperparameters unchanged. These tuned values were then applied to DiT-Aug.

### C.3 GEOM-DRUGS DETAILS

For all of our experiments, we retain the diffusion hyperparameters as EDM (Hoogeboom et al., 2022) - i.e. we use the same noise schedule, discretisation steps etc.

Like with QM9, we report the seconds per epoch, sampling time (s) and vRAM for the models used in Table 4. We exclude END as their code is not publicly available. We omit the results for EDM and GeoLDM as were unable to run their code on our NVIDIA H100 80GB GPU.

Table 7: Seconds per epochs, sampling time, and vRAM for different models on GEOM-Drugs.

| Method | # Parameters | Sec./epoch | Sampling time (s) | vRAM (GB) |
|---|---|---|---|---|
| GeoLM | 5.5M | – | – | – |
| EDM | 2.4M | – | – | – |
| SymDiff | 31.2M | 4336.82 | 0.39 | 63 |

For the SymDiff model, we used the same hyperparameters as for QM9 except for the learning where we used $10^{-4}$ as we found this to result in a lower validation loss.

### C.4 PRETRAIN-FINETUNING

To further explore the flexibility of our approach, we experimented with using it in the pretrain-finetune framework, similar to Mondal et al. (2023). Using QM9, we took the trained DiT model from Table 1 and substituted it as the $\epsilon_\theta$ for our SymDiff model, while keeping the same architecture and hyperparameters for $f_\theta$. We tested two setups: finetuning both $\epsilon_\theta$ and $f_\theta$ (DiT-FT) and freezing $\epsilon_\theta$ while tuning only $f_\theta$ (DiT-FT-Freeze). The same training procedure and optimization hyperparameters were used, except we now trained our models for only 800 epochs and used a larger grid for learning rate and weight decay tuning. Specifically, we searched first for the optimal learning

rate in $[10^{-3}, 8 \times 10^{-4}, 2 \times 10^{-4}, 10^{-4}]$ and for the optimal weight decay in $[0, 10^{-12}, 2 \times 10^{-12}]$. We found the optimal learning rate and weight decay to be $10^{-3}$ and $2 \times 10^{-12}$.

Table 8: Test NLL, atom stability, molecular stability, validity and uniqueness on QM9 for 10,000 samples and 3 evaluation runs.

| Method | NLL $\downarrow$ | Atm. stability (%) $\uparrow$ | Mol. stability (%) $\uparrow$ | Val. (%) $\uparrow$ | Uniq. (%) $\uparrow$ |
|---|---|---|---|---|---|
| SymDiff | **-129.35**$_{\pm 1.07}$ | **98.74**$_{\pm 0.03}$ | **87.49**$_{\pm 0.23}$ | **95.75**$_{\pm 0.10}$ | 97.89$_{\pm 0.26}$ |
| DiT-FT | -111.66$_{\pm 1.22}$ | 98.43$_{\pm 0.03}$ | 83.27 $_{\pm 0.39}$ | 94.19$_{\pm 0.16}$ | 98.17$_{\pm 0.26}$ |
| DiT-FT-Freeze | -43.29$_{\pm 3.73}$ | 95.68$_{\pm 0.02}$ | 55.02$_{\pm 0.38}$ | 90.48$_{\pm 0.24}$ | **99.06**$_{\pm 0.13}$ |
| DiT | -127.78$_{\pm 2.49}$ | 98.23$_{\pm 0.04}$ | 81.03$_{\pm 0.25}$ | 94.71$_{\pm 0.31}$ | 97.98$_{\pm 0.12}$ |

From Table 8, we observe that finetuning both $\epsilon_\theta$ and $f_\theta$ improves performance over the DiT model, even with our minimal tuning. However, finetuning only $f_\theta$ leads to worse results, indicating that end-to-end training or finetuning the whole model is necessary. This underscores the flexibility of our approach and its potential for easy and efficient symmetrisation of pretrained DiT models with an unconstrained $f_\theta$.

## D    DISCUSSION ABOUT THE SYMDIFF OBJECTIVE

We explain here why the SYMDIFF objective in equation 9 is reasonable to use as a surrogate for the true ELBO in equation 4. The underlying idea is analogous to Remark 7.1 of Cornish (2024). First, it is straightforward to check that our SYMDIFF objective recovers the ELBO exactly if either of the following two conditions are met:

- $\gamma_\theta$ is deterministic, i.e. $\gamma_\theta(dg|\mathbf{z}_1)$ is a Dirac distribution for every $\mathbf{z}_1 \in \mathcal{Z}$; or

- $k_\theta$ is $\mathcal{G}$-equivariant.

(For our model in Section 3.3, the latter holds if the function $\epsilon_\theta : \mathcal{Z} \to \mathcal{Z}$ is deterministically $O(3)$-equivariant.) It follows that the result of optimising our SYMDIFF objective will achieve at least as high an ELBO as the best performing $\theta$ for which either of these two conditions are met. Accordingly, if our model is powerful enough to express (or approximate) a rich family of deterministic $\gamma_\theta$ and $\mathcal{G}$-equivariant $k_\theta$, then it is reasonable to expect good performance from our surrogate objective. More generally, our model also has the ability to interpolate between these two conditions, allowing for potentially better overall optima than could be achieved in either case individually.

## E    EXTENSION TO SCORE MATCHING AND FLOW MATCHING

In this section, we discuss how to extend stochastic symmetrisation to score and flow-based generative models to give an analogue of SYMDIFF to these paradigms. For clarity of presentation, we consider all models to be defined for $N$-body systems living in the full space $\mathcal{Z} = \mathbb{R}^{N \times 3}$ where we wish to obtain a $S_N \times O(3)$-equivariant model - i.e. we do not consider non-geometric features or translation invariance. Although, we note that the below discussion can be extended to such settings in the natural way as presented for diffusion models above.

### E.1    SCORE MATCHING

Score-based generative models (SGMs) (Song et al., 2020) are the continuous-time analogue of diffusion models. SGMs consider the forward noising process $\mathbf{x}_t \sim p_t$ for $t \in [0, T]$ defined by the following stochastic differential equation (SDE) with the initial condition $\mathbf{x}_0 \sim p_{\text{data}}$:

$$d\mathbf{x}_t = f(\mathbf{x}_t, t)dt + g(t)d\mathbf{w}, \tag{22}$$

for some choice of functions $f : \mathcal{Z} \times [0, T] \to \mathcal{Z}$ and $g : [0, T] \to \mathbb{R}$, and where $\mathbf{w}$ is a standard Weiner process.

The corresponding backward process is shown in Anderson (1982) to take the form:

$$d\mathbf{x}_t = \left[ f(\mathbf{x}_t, t) - g(t)^2 \nabla_\mathbf{x} \log p_t(\mathbf{x}_t) \right] dt + g(t) d\bar{\mathbf{w}}, \tag{23}$$

where $\bar{\mathbf{w}}$ is a standard Weiner process and time runs backwards from $T$ to $0$. Hence, given samples $\mathbf{x}_T \sim p_T$ and access to the score of the marginal distributions $\nabla_\mathbf{x} \log p_t(\mathbf{x}_t)$, we can obtain samples from $p_{\text{data}}$ by simulating the backward process in equation 23.

By considering the Euler–Maruyama discretisation of equation 23, we can represent the sampling scheme of a SGM in terms of the Markov chain $p_T(\mathbf{x}_T) \prod_{i=1}^n p(\mathbf{x}_{t_{i-1}} | \mathbf{x}_{t_i})$, where the time-points $t_i$ are uniformly spaced in $[0, T]$ - i.e. $t_i = i\Delta t$ where $\Delta t = T/n$ - and the reverse transition kernels are given by:

$$p(\mathbf{x}_{t_{i-1}} | \mathbf{x}_{t_i}) = \mathcal{N} \left( \mathbf{x}_{t_{i-1}}; \mathbf{x}_{t_i} + \Delta t \left\{ f(\mathbf{x}_{t_i}, t_i) - g(t_i)^2 \nabla_x \log p_{t_i}(\mathbf{x}_{t_i}) \right\}, g(t_i)^2 \Delta t \mathbf{I} \right).$$

In what follows, we additionally assume that $f(\cdot, t)$ is $S_N \times \mathrm{O}(3)$-equivariant for all $t \in [0, T]$. This is true for common choices of $f$ which take $f$ to be linear in $\mathbf{x}_t$.

**Stochastic symmetrisation** In order to learn an approximation to the transition kernels via stochastic symmetrisation, we can parametrise the reverse transition kernels, in a similar fashion as for diffusion models, by

$$p_\theta(\mathbf{x}_{t_{i-1}} | \mathbf{x}_{t_i}) = \mathsf{sym}_{\gamma_\theta}(k_\theta)(\mathbf{x}_{t_{i-1}} | \mathbf{x}_{t_i}),$$

where we take $k_\theta(\mathbf{x}_{t_{i-1}} | \mathbf{x}_{t_i}) = \mathcal{N}(\mathbf{x}_{t_{i-1}}; \mu_\theta(\mathbf{x}_{t_i}), g(t_i)^2 \Delta t \mathbf{I})$. We define $\mu_\theta$ by the following parametrisation[5]:

$$\mu_\theta(\mathbf{x}_{t_i}) = \mathbf{x}_{t_i} + \Delta t \left\{ f(\mathbf{x}_{t_i}, t_i) - g(t_i)^2 s_\theta(\mathbf{x}_{t_i}) \right\},$$

where we take $s_\theta : \mathcal{Z} \to \mathcal{Z}$ to be a $S_N$-equivariant neural network which aims to learn an approximation to the true score $\nabla_\mathbf{x} \log p_t(\mathbf{x}_t)$. This ensures that $k_\theta$ is $S_N$-invariant. Additionally, we assume that $\gamma_\theta : \mathcal{Z} \to \mathrm{O}(3)$ is some choice of a $S_N$-invariant and $\mathrm{O}(3)$-equivariant Markov kernel. Hence, we can conclude that $p_\theta : \mathcal{Z} \to \mathcal{Z}$ is a $S_N \times \mathrm{O}(3)$-equivariant Markov kernel by Theorem 1. We can also guarantee that $p_\theta$ admits a density by Proposition 1.

**Training** To learn $\theta$ for $p_\theta(\mathbf{x}_{t_{i-1}} | \mathbf{x}_{t_i})$, a natural objective is to minimise the KL divergence between the true reverse kernels and our parametrised reverse kernels

$$\mathcal{L}(\theta) = \sum_{i=1}^n \lambda_0(t_i) \mathcal{L}_i(\theta), \qquad \mathcal{L}_i(\theta) = \mathbb{E}_{p_{t_i}(\mathbf{x}_{t_i})} \left[ D_{\mathrm{KL}}(p(\mathbf{x}_{t_{i-1}} | \mathbf{x}_{t_i}) || p_\theta(\mathbf{x}_{t_{i-1}} | \mathbf{x}_{t_i})) \right],$$

where $\lambda_0$ is some time weighting function. We note that we run into the same issue as with SYMDIFF in that we do not have access to $p_\theta(\mathbf{x}_{t_{i-1}} | \mathbf{x}_{t_i})$ in closed-form, since this is expressed in terms of an expectation. However, as $\mathcal{L}_i(\theta)$ is a linear function of $-\log p_\theta(\mathbf{x}_{t_{i-1}} | \mathbf{x}_{t_i})$ and $-\log$ is a convex function, we can apply Jensen's inequality again to provide the following upper bound to our original objective

$$\mathcal{L}'(\theta) = \sum_{i=1}^n \lambda_0(t_i) \mathcal{L}'_i(\theta), \qquad \mathcal{L}'_i(\theta) = \mathbb{E}_{p_{t_i}(\mathbf{x}_{t_i}), \gamma_\theta(dR|\mathbf{x}_{t_i})} \left[ D_{\mathrm{KL}}(p(\mathbf{x}_{t_{i-1}} | \mathbf{x}_{t_i}) || k_\theta(\mathbf{x}_{t_{i-1}} | R, \mathbf{x}_{t_i})) \right],$$

which we can now use to train $\theta$. To further simplify $\mathcal{L}'_i(\theta)$, we note that $k_\theta(\mathbf{x}_{t_{i-1}} | R, \mathbf{x}_{t_i}) = \mathcal{N}(\mathbf{x}_{t_{i-1}}; R \cdot \mu_\theta(R^T \cdot \mathbf{x}_{t_i}), g(t_i)^2 \Delta t \mathbf{I})$ with a similar derivation as before. This allows us to evaluate the KL divergences in closed form since $p, k_\theta$ are defined in terms of Gaussians. We can show that this gives

$$\mathcal{L}'_i(\theta) = \mathbb{E}_{p_{t_i}(\mathbf{x}_{t_i}), \gamma_\theta(dR|\mathbf{x}_{t_i})} \left[ \frac{1}{2} g(t_i)^2 \Delta t \big\| R \cdot s_\theta(R^T \cdot \mathbf{x}_{t_i}) - \nabla_\mathbf{x} \log p_{t_i}(\mathbf{x}_{t_i}) \big\|^2 \right], \tag{24}$$

where we use the fact that $f(\cdot, t)$ is $S_N \times \mathrm{O}(3)$-equivariant. To express equation 24 in a tractable form (as we do not have access to the true score), we can apply the standard technique of employing the score matching identity (Vincent, 2011) to give

$$\mathcal{L}'_i(\theta) = \mathbb{E}_{p(\mathbf{x}_0), p(\mathbf{x}_{t_i} | \mathbf{x}_0), \gamma_\theta(dR|\mathbf{x}_{t_i})} \left[ \frac{1}{2} g(t_i)^2 \Delta t \big\| R \cdot s_\theta(R^T \cdot \mathbf{x}_{t_i}) - \nabla_\mathbf{x} \log p_{t_i}(\mathbf{x}_{t_i} | \mathbf{x}_0) \big\|^2 \right] + C_i,$$

---

[5] Similar to our discussion on diffusion models, we leave the time dependency implicit in here.

where $p(\mathbf{x}_{t_i}|\mathbf{x}_0)$ denotes the conditional distribution of $\mathbf{x}_{t_i}$ given $\mathbf{x}_0$ under the forward noising process $p$, and $C_i$ is some constant. In practice, the choice of forward noising SDE in equation 22 is made to ensure that we have access to $p(\mathbf{x}_{t_i}|\mathbf{x}_0)$ in closed-form and that the distribution is easy to sample from.

By making the choice that $\lambda_0(t) = 2\lambda(t)/(g(t_i)^2 T)$ for some suitable time weighting function $\lambda$, we can show as $\Delta t \to 0$ that our objective $\mathcal{L}'(\theta)$ will converge to (modulo some constant)

$$\mathbb{E}_{p(\mathbf{x}_0),t\sim U(0,T),p(\mathbf{x}_t|\mathbf{x}_0),\gamma_\theta(dR|\mathbf{x}_t)} \left[ \lambda(t) \big\| R \cdot s_\theta(R^T \cdot \mathbf{x}_t) - \nabla_\mathbf{x} \log p(\mathbf{x}_t|\mathbf{x}_0) \big\|^2 \right], \qquad (25)$$

where $U(0,T)$ denotes the uniform distribution on $[0,T]$. We see that our final objective in equation 25 now resembles the standard score matching objective.

### E.2 FLOW MATCHING

Continuous normalising flows (CNFs) (Chen et al., 2018) construct a generative model of data $\mathbf{x}_1 \sim q = p_{\text{data}}$ by the pushforward of an ordinary differential equation (ODE) taking the form

$$\frac{d}{dt}\phi_t(\mathbf{x}) = u_t(\phi_t(\mathbf{x})), \qquad \phi_0(\mathbf{x}) = \mathbf{x}, \qquad (26)$$

where $u_t : \mathcal{Z} \times [0,T] \to \mathcal{Z}$ is the vector field function defining the ODE, and $\phi_t : \mathcal{Z} \times [0,T] \to \mathcal{Z}$ denotes the flow implicitly defined by solutions to the above ODE. By letting $p_0$ be some simple prior distribution, the above ODE defines a generative model $\mathbf{x}_t \sim p_t$ by the pushforward of $p_0$ through the flow $\phi_t$

$$p_t = [\phi_t]_\# p_0 \qquad (27)$$

If $u_t$ is chosen in such a way that $p_1 \approx q = p_{\text{data}}$, we can then generate samples from $p_{\text{data}}$ by sampling some $\mathbf{x}_0 \sim p_0$, then solving the ODE in equation 26 with this initial condition[6]. Furthermore, as in previous work (Klein et al., 2024), we assume that $u_t$ is $S_N \times O(3)$-equivariant.

By considering the Euler discretisation of equation 26, we can represent the generation process by the Markov chain $p_0(\mathbf{x}_0) \prod_{i=1}^T p(d\mathbf{x}_{t_i}|\mathbf{x}_{t_{i-1}})$ where the time-points $t_i$ are uniformly spaced in $[0,T]$ - i.e. $t_i = i\Delta t$ where $\Delta t = T/n$ - and the transition kernels are given by the Markov kernels

$$p(d\mathbf{x}_{t_i}|\mathbf{x}_{t_{i-1}}) = \delta(\mathbf{x}_{t_{i-1}} + u_{t_{i-1}}(\mathbf{x}_{t_{i-1}})\Delta t), \qquad (28)$$

where $\delta(\cdot)$ denotes the Dirac measure at some point.

**Stochastic symmetrisation** To learn the transition kernels induced by the vector field $u_t$ via stochastic symmetrisation, we parametrise our transition kernels by

$$p_\theta(d\mathbf{x}_{t_i}|\mathbf{x}_{t_{i-1}}) = \text{sym}_{\gamma_\theta}(k_\theta)(\mathbf{x}_{t_i}|\mathbf{x}_{t_{i-1}}), \qquad (29)$$

where we take $k_\theta(d\mathbf{x}_{t_i}|\mathbf{x}_{t_{i-1}}) = \delta(\mathbf{x}_{t_{i-1}} + v_{t_{i-1}}^\theta(\mathbf{x}_{t_{i-1}})\Delta t)$ in which $v_t^\theta : \mathcal{Z} \to \mathcal{Z}$ is some $S_N$-equivariant neural network which aims to learn an approximation to the true vector field $u_t$. We further assume $\gamma_\theta : \mathcal{Z} \to O(3)$ is some $S_N$-invariant and $O(3)$-equivariant Markov kernel. We can again conclude that $p_\theta : \mathcal{Z} \to \mathcal{Z}$ is a $(S_N \times O(3))$-equivariant Markov kernel by Theorem 1.

**Training** A natural objective to learn $p_\theta$ is to minimise the 2-Wasserstein distance $\mathcal{W}_2$ (Peyré et al., 2019) between $p(d\mathbf{x}_{t_i}|\mathbf{x}_{t_{i-1}})$ and $p_\theta(d\mathbf{x}_{t_i}|\mathbf{x}_{t_{i-1}})$ since these are defined in terms of Dirac measures. We can write our objective as

$$\mathcal{L}(\theta) = \sum_{i=1}^T \lambda_0(t_{i-1})\mathcal{L}_i(\theta), \qquad \mathcal{L}_i(\theta) = \mathbb{E}_{p_{t_{i-1}}(\mathbf{x}_{t_{i-1}})} \left[ \mathcal{W}_2^2(p(d\mathbf{x}_{t_i}|\mathbf{x}_{t_{i-1}}), p_\theta(d\mathbf{x}_{t_i}|\mathbf{x}_{t_{i-1}})) \right]$$

where $\lambda_0$ is some time weighting function, and the 2-Wasserstein distance $\mathcal{W}_2$ is defined as $\mathcal{W}_2^2(\pi_1, \pi_2) = \inf_\pi \int \|\mathbf{x} - \mathbf{y}\|^2 \, d\pi(\mathbf{x}, \mathbf{y})$ where $\pi$ is taken over the space of possible couplings

---

[6]The use of time here reverse the convention used in the diffusion literature.

between the measures $\pi_1, \pi_2$. We note that as $p(d\mathbf{x}_{t_i}|\mathbf{x}_{t_{i-1}})$ is Dirac, there only exists a single coupling between the two kernels given by the product of the Markov kernels. This allows us to evaluate $\mathcal{L}_{i+1}$ as

$$\mathcal{L}_{i+1}(\theta) = \mathbb{E}_{p_{t_i}(\mathbf{x}_{t_i}), \gamma_\theta(dR|\mathbf{x}_{t_i})} \left[ \Delta t^2 \big\| R \cdot v_{t_i}^\theta(R^T \cdot \mathbf{x}_{t_i}) - u_{t_i}(\mathbf{x}_{t_i}) \big\|^2 \right]. \tag{30}$$

To express equation 30 in a tractable form (as we do not have access to $u_t$), we can take $u_t$ to be constructed by the same setup used in Flow Matching (Lipman et al., 2022). This framework allows us to express equation 30 in the now tractable form

$$\mathcal{L}_{i+1}(\theta) = \mathbb{E}_{q(\mathbf{x}_1), p_{t_i}(\mathbf{x}_{t_i}|\mathbf{x}_1), \gamma_\theta(dR|\mathbf{x}_{t_i})} \left[ \Delta t^2 \big\| R \cdot v_{t_i}^\theta(R \cdot \mathbf{x}_{t_i}) - u_{t_i}(\mathbf{x}_{t_i}|\mathbf{x}_1) \big\| \right] + C_{i+1}, \tag{31}$$

by the use of the Conditional Flow Matching objective, where $C_{i+1}$ is some constant. Here $p_t(\mathbf{x}_t|\mathbf{x}_1)$ is a family of conditional distributions where $p_0(\mathbf{x}_0|\mathbf{x}_1) = p_0(\mathbf{x}_0)$ equals our prior distribution and $p_1(\mathbf{x}_1|\mathbf{x}_1) \approx \delta(\mathbf{x}_1)$, and for which $u_t(\mathbf{x}_t|\mathbf{x}_1)$ is a vector field generating $p_t(\mathbf{x}_t|\mathbf{x}_1)$ by an ODE of the form in equation 27. These are constructed to be easy to sample from and evaluate. The true vector field $u_t$, which provides a generative model of $q = p_{\text{data}}$, is then defined by some expectation of the conditional vector fields $u_t(\mathbf{x}_t|\mathbf{x}_0)$ over $p_t(\mathbf{x}_t|\mathbf{x}_0)$ and $q(\mathbf{x}_1)$.

Hence, by taking $\lambda_0(t) = \frac{\lambda(t)}{T \Delta t}$ for some suitable time weighting function $\lambda$, we can show as $\Delta t \to 0$, our objective $\mathcal{L}(\theta)$ will converge to (modulo some constant)

$$\mathbb{E}_{q(\mathbf{x}_1), t \sim U(0,T), p_t(\mathbf{x}_t|\mathbf{x}_1), \gamma_\theta(dR|\mathbf{x}_t)} \left[ \lambda(t) \big\| R \cdot v_t^\theta(R^T \cdot \mathbf{x}_t) - u_t(\mathbf{x}_t|\mathbf{x}_1) \big\|^2 \right]. \tag{32}$$

We see that our final objective in equation 32 now resembles the standard flow matching objective.

