# OpenReview forum: "SymDiff: Equivariant Diffusion via Stochastic Symmetrisation"
_ICLR.cc/2025/Conference — ICLR 2025 Poster_

### Official Review · Reviewer_GD6U · 2024-11-01

**Soundness:** 3
**Presentation:** 3
**Contribution:** 3
**Rating:** 5
**Confidence:** 5

**Summary:**

This paper proposes SymDiff, a novel method for constructing equivariant diffusion models via stochastic symmetrisation. It addresses limitations of prior work on geometric data by allowing more flexibility than using intrinsically equivariant neural networks. Experiments on QM9 and GEOM-Drugs datasets for molecular generation show SymDiff outperforms EDM and is competitive with other baselines, with better scalability. SymDiff can be seen as a generalisation of data augmentation, with a learned and deployed augmentation process ensuring stochastic equivariance.

**Strengths:**

- Originality: The paper proposes SymDiff, a novel method using stochastic symmetrisation for equivariant diffusion models, extending symmetrisation to generative modelling and differing from prior approaches.

- Clarity: The paper is well-structured, clearly explaining concepts with examples and presenting algorithms, aiding understanding and implementation.

- Quality: The paper presents a comprehensive framework with solid derivations and conducts rigorous experiments on multiple datasets.

**Weaknesses:**

- Performance gap: from Tab2 and 3, seems the model is not parameter efficient compared with baseline models. When reducing the model size, the performance will have a larger drop than the baseline model.

- Unfair comparison: I didn't fully understand for SymDiff+++, ++, and +, why you cannot choose the same number of parameters of EDM. It seems your model SymDiff+ is already larger than EDM+++, but with worse performance.

- Missing comparison: I think the author also needs to incorporate GeoLDM with different sizes into Tab 2 and 3, which potentially can work better since it acts on low-dimensional space.

**Questions:**

- Why SymDiff require larger model size to match the performance of EDM / GeoLDM etc.

---

> ### Author Response · Authors · 2024-11-19
> **Response to reviewer GD6U**
>
> Thanks very much for your review, and we are glad to hear that you found the paper clear and high quality.
> Please see responses to your individual points below.
>
> **Second point**
>
> > Unfair comparison: I didn't fully understand for SymDiff+++, ++, and +, why you cannot choose the same number of parameters of EDM. It seems your model SymDiff+ is already larger than EDM+++, but with worse performance.
>
> Please see our overall response to all reviewers above.
> In summary, we believe that overall computational cost (computational time and vRAM) is what ultimately matters in practice, and since the neural net architectures here are so different (GNN vs. Transformer), parameter counts do not give a good measure of this.
> We have provided additional benchmarks above that demonstrate our method yields the same or improved performance at a lower computational cost.
>
> **First point**
>
> > Performance gap: from Tab2 and 3, seems the model is not parameter efficient compared with baseline models.
> When reducing the model size, the performance will have a larger drop than the baseline model.
>
> Thanks for pointing this out.
> We think that Table 2 and 3 were confusing, and have now streamlined into Table 3 in the updated manuscript. For convenience, we have copied Table 3 below.
>
> | Method | NLL $\downarrow$ | Mol. stability (\%) $\uparrow$ | Sec./epoch (s) $\downarrow$ | Sampling time (s) $\downarrow$ | vRAM (GB) $\downarrow$ |
> | --- | --- | --- | --- | --- | --- |
> | EDM$^{++}$ | -119.12$\pm{1.41}$ | 85.68$\pm{0.83}$ | 160.60 | 0.56 | 23 |
> | EDM$^{+}$ | -110.97$\pm{1.42}$ | 84.63$\pm{0.16}$ | 192.60 | 0.46 | 23 |
> | EDM  | -110.70$\pm{1.50}$ | 82.00 $\pm{0.40}$ | 88.80 | 0.27 | 14 |
> | SymDiff  | -129.35$\pm{1.07}$ | 87.49$\pm{0.23}$ | 27.20 | 0.09 | 7 |
>  | SymDiff$^{-}$ |  -125.40$\pm{0.63}$ | 83.51$\pm{0.24}$ | 24.87 | 0.08 | 6 |
> | SymDiff$^{--}$  | -110.68 $\pm{2.55}$ | 71.25 $\pm{0.50}$ | 20.60 | 0.07 | 5 |
>
> The key point here is the following:
> * The SymDiff model from Table 1 is the *largest* SymDiff model in Table 2 and 3.
> * The EDM model from Table 1 is the *smallest* EDM model from Table 2 and 3.
>
> In effect, these tables show the result of making the SymDiff model from Table 1 smaller, and the EDM model bigger. In essence, we have
> $$
>     \text{SymDiff}^{--} \prec \text{SymDiff}^{-} \prec \text{SymDiff} \prec \text{EDM} \prec \text{EDM}^{+} \prec \text{EDM}^{++}.
> $$
> where $\prec$ denotes smaller computational cost and we have renamed the models compared to the first version as follows:
> \begin{align}
>     &\text{SymDiff}^{+++} \to \text{SymDiff}^{}, \ \text{SymDiff}^{++} \to \text{SymDiff}^{-}, \ \text{SymDiff}^{+} \to \text{SymDiff}^{--}, \\
>     &\text{EDM}^{+++} \to \text{EDM}^{++}, \ \text{EDM}^{++} \to \text{EDM}^{+}, \ \text{EDM}^{+} \to \text{EDM}^{}
> \end{align}
> So we are not comparing like-for-like: the smallest SymDiff model here (SymDiff--) is much smaller in terms of computational cost than the smallest EDM model (EDM).
>
> From this perspective, SymDiff again is more computationally efficient.
> From the updated version of Table 3, SymDiff$^{-}$ *still* does better than EDM in terms of molecular stability, but has considerably lower computational cost.
> Likewise, SymDiff does better than even the largest EDM model, EDM$^{++}$, again even with considerably lower cost.

---

> > ### Author Response · Authors · 2024-11-20
> > **Continued response to reviewer GD6U**
> >
> > **Third point**
> >
> > > Missing comparison: I think the author also needs to incorporate GeoLDM with different sizes into Tab 2 and 3, which potentially can work better since it acts on low-dimensional space.
> >
> > We have included additional benchmarking information for GeoLDM in the overall response to all reviewers. Overall, GeoLDM performs roughly on par with SymDiff$^*$ in terms of molecular stability, but is much more computationally costly.
> > Therefore, to ensure a fair comparison, we would also need to scale up SymDiff$^*$ to compare against a larger GeoLDM.
> > (Please also see our response to the last point above: really, Tables 2 and 3 were showing the result of making SymDiff *smaller*, rather than bigger.)
> >
> > We also agree that the low-dimensional latent space of GeoLDM seems to be very effective in that paper.
> > We note that, being a very flexible method, SymDiff could be used in conjunction with an autoencoder approach like this (as opposed to a simple DiT as we used). This would potentially yield further performance benefits beyond those we reported over EDM, which was our most direct point of comparison.
> >
> > Potentially the performance metrics for GeoLDM compared with SymDiff$^{*}$ address your concerns already. If not, please let us know, and we will try to run additional comparisons in time for the deadline as well.
> >
> > **Conclusion**
> >
> > Please let us know if we provide any additional information or clarifications.
> > If we have adequately addressed your points, we hope you will consider increasing your score.

---

### Official Review · Reviewer_RgSJ · 2024-11-03

**Soundness:** 4
**Presentation:** 4
**Contribution:** 4
**Rating:** 8
**Confidence:** 3

**Summary:**

The paper “SymDiff: Equivariant Diffusion via Stochastic Symmetrisation” introduces SYMDIFF, a novel method for constructing equivariant diffusion models without the need for complex, intrinsically equivariant neural networks. Traditional approaches often rely on specialized architectures with complex parameterizations and higher-order geometric features to achieve equivariance, which can be computationally intensive and challenging to implement.

SYMDIFF leverages the framework of stochastic symmetrisation, effectively functioning as a learned data augmentation technique applied during the sampling phase. This approach is lightweight, computationally efficient, and can be easily implemented on top of arbitrary off-the-shelf models. By not requiring the neural network components to be intrinsically equivariant, SYMDIFF allows the use of highly scalable modern architectures, providing greater flexibility and performance.

The authors demonstrate the effectiveness and scalability of SYMDIFF through empirical experiments on E(3)-equivariant molecular generation tasks using the well-known QM9 and GEOM-DRUGS datasets. The results show significant improvements over traditional methods, highlighting the benefits of SYMDIFF in generating molecular structures that are equivariant under the E(3) group (which includes translations, rotations, and reflections in three-dimensional space).

This work represents the first application of stochastic symmetrisation to generative modeling, suggesting that SYMDIFF has the potential for broader applications in the domain of equivariant machine learning models.

**Strengths:**

This paper introduces SYMDIFF, a simple yet powerful method for constructing equivariant diffusion models without relying on complex intrinsically equivariant neural network architectures. The strengths of the paper can be assessed across the following dimensions:

Originality:
The paper presents a novel application of stochastic symmetrisation to generative modeling, specifically in the context of diffusion models. This approach is original because it departs from traditional methods that require specialized equivariant architectures with complex parameterizations and higher-order geometric features. By leveraging stochastic symmetrisation as a learned data augmentation technique applied during sampling, the paper creatively combines existing ideas to overcome limitations of prior models. This opens up new possibilities for using standard neural network architectures in equivariant settings.

Quality:
The paper demonstrates high quality in both theoretical and empirical aspects. It provides a solid theoretical foundation based on Markov kernels, offering a clear understanding of how SYMDIFF achieves equivariance without specialized architectures. The empirical evaluations are thorough, showcasing the method’s effectiveness on E(3)-equivariant molecular generation tasks using the well-known QM9 and GEOM-DRUGS datasets. The results indicate significant performance improvements and scalability advantages over traditional equivariant models. The experiments are well-designed, and the conclusions are well-supported by the data.

Clarity:
The paper is written with clarity and precision, making complex concepts accessible to the reader. It systematically introduces the theoretical background before delving into the methodological details of SYMDIFF. The explanations are clear, and algorithmic descriptions helps in understanding the implementation. The empirical results are presented in an organized manner, with insightful discussions that highlight the implications of the findings. Overall, the paper effectively communicates its contributions and significance.

Significance:
The significance of the paper lies in its potential to impact the field of equivariant machine learning models substantially. By eliminating the need for specialized equivariant architectures, SYMDIFF lowers the barrier to entry for researchers and practitioners interested in building equivariant models. It enables the use of arbitrary off-the-shelf neural network architectures, which are often more scalable and easier to implement. This flexibility can accelerate the development and deployment of equivariant models in various applications, including molecular generation, computer vision, and physics-based simulations. Moreover, being the first to apply stochastic symmetrisation to generative modeling, the paper paves the way for future research in this promising direction.

**Weaknesses:**

While the paper introduces SYMDIFF as a flexible and efficient method for constructing equivariant diffusion models, it lacks specific guidelines to help practitioners decide when to use SYMDIFF over traditional intrinsically equivariant models. This absence of decision-making support makes it challenging to understand the circumstances under which SYMDIFF is more advantageous.

To address this, I suggest the authors include a comparative analysis between SYMDIFF and traditional methods. This could be presented as a table or flowchart highlighting key factors such as:

- Dataset Characteristics: Situations where SYMDIFF performs better or worse.
- Model Complexity: Differences in architecture and scalability.
- Computational Resources: Comparisons of memory usage, training time, and inference speed.
- Performance Metrics: Quantitative comparisons on accuracy and sample quality.
- Ease of Implementation: Practical considerations like code complexity.

Including such a comparison would provide clear guidance for practitioners, making the paper more actionable and useful.

Additionally, while the paper claims that SYMDIFF is computationally efficient, it doesn’t provide empirical evidence to support this. I recommend the authors include:

- Empirical Benchmarks: Runtime analysis and resource consumption statistics compared to intrinsically equivariant models.
- Convergence Plots: Performance versus number of training epochs for both SYMDIFF and baseline methods.
- Ablation Studies: Impact of stochastic symmetrisation on computational overhead and model performance.

By incorporating these elements, the paper would strengthen its claims and offer actionable insights, enabling readers to make informed decisions about adopting SYMDIFF.

**Questions:**

- Do you have insights into how many training epochs are necessary for SYMDIFF to effectively learn equivariance in the molecular generation task?
- How does the convergence speed of SYMDIFF compare to that of intrinsically equivariant models in learning equivariant properties?
- In the context of molecular generation, do you consider the equivariant component of the task to be a minor or major challenge compared to learning the underlying chemical laws necessary for accurate atom placement?

---

> ### Author Response · Authors · 2024-11-19
> **Response to reviewer RgSJ**
>
> Thank you very much for the positive feedback and review score. We are glad that you found our paper to be well-written and useful to the research community. Please see below for our response to the points raised within your weaknesses and questions section.
>
> **Further comparisons with SymDiff**
>
> > I suggest the authors include a comparative analysis between SYMDIFF and traditional methods ... Additionally, while the paper claims that SYMDIFF is computationally efficient, it doesn’t provide empirical evidence to support this ...
>
> Please see our overall response to all the reviewers above for an extended comparison between SymDiff and the other methods that we benchmark against in the paper.
>
> **Miscellaneous questions**
>
> > Do you have insights into how many training epochs are necessary for SYMDIFF to effectively learn equivariance in the molecular generation task?
>
> The model used in SymDiff is (stochastically) equivariant by construction (Proposition 1 in the paper).
> The flexibility of our method is not that the model can learn to enforce equivariance to a greater or lesser extent, but that we have more architectural freedom within our model. This is because the equivariance in SymDiff does not require any intrinsically equivariant components. We show in the overall response above that this leads to a reduction in computational complexity while either maintaining or increasing performance compared to other methods which lack this architectural freedom.
>
> > How does the convergence speed of SYMDIFF compare to that of intrinsically equivariant models in learning equivariant properties?
>
> From above, the model does not learn to enforce equivariance as this is already enforced by construction.
> However, we can consider the convergence rate of the models in terms of the metrics we use.
> For example, we observed that the convergence rate of $\text{SymDiff}$ (31M) in terms of molecular stability was comparable to the baseline EDM model and was faster when comparing against the larger variants of EDM from Table 3 in our paper.
>
> > In the context of molecular generation, do you consider the equivariant component of the task to be a minor or major challenge compared to learning the underlying chemical laws necessary for accurate atom placement?
>
> Equivariance is an important inductive bias for handling geometric structure in molecular generation tasks as it provides a principled way to reduce the complexity of the generation task by processing the symmetries of molecules in a systematic fashion. However, equivariance can only handle global symmetries within the geometry and cannot directly inform the model about the chemical laws which are the main factor determining the structure and function of molecules. Hence, enforcing equivariance is an important yet less difficult challenge compared to learning the fundamental chemical laws which underlie molecular structure.

---

### Official Review · Reviewer_u6Xc · 2024-11-03

**Soundness:** 3
**Presentation:** 3
**Contribution:** 3
**Rating:** 8
**Confidence:** 4

**Summary:**

This paper proposes a new way to build equivariant diffusion models. Whereas traditional methods rely either on simple data augmentation strategies or neural network architectures that are directly constructed to be equivariant, *SymDiff* uses stochastic symmetrization to learn equivariant diffusion models. Stochastic symmetrization is a recently introduced framework by Cornish (2024). On a high level, stochastic symmetrization can intuitively be seen as an advanced data augmentation method, where the augmentation is sampled from an additional, learnable Markov kernel and where the main denoiser network does not need to be equivariant itself. Moreover, the augmentation is also applied during sampling. The paper in detail introduces the approach and rigorously shows that it is principled. SymDiff is then validated on molecule generation, where the permutation equivariance is directly built into the employed networks, whereas rotation equivariance is enforced through stochastic symmetrization. The paper compares to various baselines, showing favourable results, and runs many appropriate and insightful ablation experiments.

**Strengths:**

**Clarity**: The paper is overall well written and the necessary background is introduced in an appropriate manner, such that one can follow the overall paper. However, the stochastic symmetrization framework is introduced in a fairly abstract manner and I believe more intuitions could be discussed and examples given, which would likely help the broad ICLR audience (see questions below).

**Originality**: Building equivariant diffusion models with stochastic symmetrization is an overall novel and original idea. Methodologically, the approach directly builds on the stochastic symmetrization framework from Cornish (2024). Hence, the novelty here lies in the application of the framework to diffusion models, specifically those for small molecule generation, a popular and important benchmark task.

**Experimental Results**: Overall, the paper is able to show favourable results compared to various baselines, thereby successfully validating the method, and I liked the many ablation experiments that were run.

**Quality and Significance**: Overall, this paper is of high quality and tackles an important problem. Enforcing equivariances and invariances in generative models can be crucial in applications with symmetries, as commonly encountered in scientific applications. However, it is difficult to directly build high-performance and scalable equivariant network architectures. Although quite related to data augmentation methods, stochastic symmetrization and SymDiff proposes a new way to address the problem, and I believe this paper has the potential to result in follow-up work. Hence, I believe this is sufficiently significant for ICLR.

**Weaknesses:**

I also see some weaknesses.

- *More discussion of intuitions and more experimental analyses*: I think it would be great if the authors discussed some intuitions in more detail and analyzed their models in more detail. Specifically:
  - In practice, training SymDiff boils down to the training procedure in Algorithm 1. $\gamma_\theta$ is trained as part of the ELBO objective, using reparametrization. What sort of $\gamma_\theta$ and $f_\theta$ functions are we intuitively expected to learn in different situations? How does the model intuitively exploit its flexibility to produce more complex $R$ distributions compared to when only using the plain Haar measure? Can examples and intuitions be given?
  - Moreover, is there anything preventing the model from simply learning $R = R_0 f_\theta(...) = I$, this is, no ''augmentation'' at all? In other words, could $f_\theta$ learn to ''undo'' the $R_0$ drawn from the Haar measure so as to remove any rotations? In that case, the training would reduce to standard training without the rotation equivariance, which may be beneficial from the objective's perspective considering that the network itself isn't equivariant? But this defeats the purpose of the method. Such potential edge cases should be discussed.
  - To this end, can the authors maybe show and analyze what sort of $\gamma_\theta$ distributions SymDiff ends up learning in practice?
  - The approach uses $\gamma_\theta$ (or $f_\theta$) also during sampling. What is the intuition for that? And what would happen if we didn't apply $\gamma_\theta$ during sampling? This would be an interesting discussion or ablation. Moreover, in the case without learnable $\gamma_\theta$, i.e. when using only the Haar measure, I would expect that the model would still perform fine even if no $R$ is applied during sampling.

- *Relation to data augmentation*: Related to the last point above, overall, it seems that SymDiff can be seen as a sophisticated data augmentation strategy (as also pointed out by the authors). However, the authors carefully distinguish the case between training with data augmentation (DiT-Aug) and SymDiff with Haar measure sampling only (SymDiff-H), pointing out that the only difference in that case is that SymDiff-H also applies augmentations during sampling, if I understood things correctly. However, in all experiments the performance of DiT-Aug and SymDiff-H is extremely close together. This makes me believe the application of the augmentation during sampling isn't necessary for SymDiff-H (also see point above), and that SymDiff-H really just corresponds to standard data augmentation. Related to that, it is not sufficiently well discussed, explained and analyzed how exactly this situation *qualitatively* changes when $\gamma_\theta$ is learnt (see point above).

- *Experimental results*: The performance gains over the EDM baseline are meaningfully large, but at the same time compared to more sophisticated baseline models SymDiff does not perform much better. Further, the authors make the point that the comparison to EDM is fair. However, the very different network sizes still make a fair comparison somewhat difficult. How would performance between EDM and SymDiff compare for similar-sized neural networks?

**Conclusion:** SymDiff is an interesting and novel method, tackling a significant problem, and I believe it is of interest to the ICLR audience. However, there are various fundamental questions about the method and the experiments (see above), that I hope the authors can answer and address. For now, I am leaning towards suggesting acceptance.

**Questions:**

I have put almost all my questions in the ''Weaknesses'' section above already, but I have one very minor additional question. In 4.2, the authors write that, given the size of GEOM-Drugs, they train a smaller model than on QM9. But the GEOM-Drugs dataset is larger and more complex than QM9. Wouldn't one want to train a larger, more expressive model in that case, and not a smaller one?

---

> ### Author Response · Authors · 2024-11-19
> **Response to reviewer u6Xc**
>
> Firstly, thanks very much for your careful reading of our paper and insightful comments and questions.
> We are very glad you found the paper high quality, interesting, and easy to follow.
>
> Please see individual responses to the points you have raised below.
>
> **More discussion of intuitions and more experimental analyses:**
>
> > What sort of $\gamma_\theta$ and $f_\theta$ functions are we intuitively expected to learn in different situations?
>
> What the model ends up learning in practice is to some extent opaque.
> However, we would argue that this is not necessarily unique to our method, but applies to most neural networks.
> Our main intuition is that:
>
> * SymDiff is always guaranteed to produce an equivariant diffusion (Proposition 1), which is known to be a good inductive bias for many applications.
> * SymDiff encompasses data augmentation as a special case (Proposition 3), which is known to be a sensible baseline.
>
> Practically speaking, our approach is then simply to make the model as large and flexible as possible, and to let it choose how it wants to actually optimise the loss.
> Assuming we optimise it well, we can then expect it to do at least as well as data augmentation, although in practice it may choose to learn something else that performs better.
> This is borne out empirically by our experiments: SymDiff did work better than data augmentation (and moreover better than intrinsically equivariant architectures).
>
> > Moreover, is there anything preventing the model from simply learning $R = R_0 f_\theta (\ldots) = I$, this is, no ''augmentation'' at all?
>
> This situation cannot occur.
> We have $R = I$ iff
> $$
>     f_\theta(R_0^T \cdot z_t, \eta) = R_0^T.
> $$
> In other words, $f_\theta$ has to learn to map $R_0^T \cdot z_t \mapsto R_0^T$.
> However, this is not possible because there are many pairs of $z_t$ and $R_0$ that map to the same $R_0^T \cdot z_t$ in general.
>
> One thing that does hold is that if the ``backbone'' of the reverse process $k_\theta(z_{t-1}|z_t)$ is *already* equivariant, then its symmetrised version is just
> $$sym_{\gamma_{\theta}} (k_{\theta})(z_{t-1}|z_{t}) = k_{\theta}(z_{t-1}|z_{t}),$$
> so that symmetrisation has no effect (Proposition 5.1 of Cornish (2024)).
> This is a good thing in terms of expressiveness, since it implies that the symmetrised model can express at least as many equivariant kernels (by choosing $\theta$ appropriately) as the backbone $k_\theta(z_{t-1}|z_t)$ can.

---

> > ### Author Response · Authors · 2024-11-19
> > **Continued response to reviewer u6Xc**
> >
> > > To this end, can the authors maybe show and analyze what sort of $\gamma_\theta$ distributions SymDiff ends up learning in practice?
> >
> > We note that the structure of the distributions learned by $\gamma_\theta$ will be dependent on the choice of architectures we use in our model.
> >
> > For the SymDiff from Table 1, please see a scatter plot of samples from $\gamma_\theta$ obtained for a fixed input at the following link: https://imgur.com/a/txgMCEQ (here we plot orthogonal matrices in $O(3)$ using the Euler angles representation).
> > This distribution appears roughly uniform, although we note that visualizing distributions over $O(3)$ is complicated by the curvature and high dimensionality of the manifold.
> > Specifically, the manifold cannot be smoothly embedded within $\mathbb{R}^3$ or $\mathbb{R}^2$.
> > Furthermore, the superior performance of the SymDiff model compared to DiT-Aug already indicates that the distributions described by $\gamma_\theta$ cannot be entirely uniform, i.e.\ it has learned something different than the Haar measure.
> >
> > In response to your question, to explore other distributions which $\gamma_\theta$ can learn in our general framework, we have also implemented a SymDiff model where, instead of the Haar, $\gamma_0$ in Equation 7 is now given by an intrinsically $S_N\times O(3)$-equivariant network using methods from [1]. Here, we kept $\gamma_1$ and $k_\theta$ to be the same DiT models as before. By Proposition 1, the resulting diffusion model will still be $S_N\times O(3)$-equivariant. We note that while this model does now depend on an intrinsically equivariant component, this part forms a very small proportion of the overall model and the rest of the model is still allowed to be parametrised freely.
> >
> > We show the distribution of the overall $\gamma_\theta$ for this model after a short training run at https://imgur.com/a/cFdMCVN for a fixed input. Compared with the previous image, the distribution is now quite different for this input, and this was also the case for other inputs as well.
> > We do not have the final trained model yet, but we will update this before the deadline.
> > However, in general, it is not clear a priori which type of behavior will be optimal.
> > Instead, this will depend in a complex way upon the training dynamics, the dataset, the backbone architecture, and so on.
> > Nevertheless, we believe that the flexibility our model has to express very different $\gamma_\theta$ distributions is a major strength of our approach.
> > This means we can express a large class of equivariant diffusion processes, and allows the model to learn whichever is optimal during training according to the specifics of the problem at hand.
> >
> > > The approach uses $\gamma_\theta$ (or $f_\theta$) also during sampling.
> > What is the intuition for that?
> > And what would happen if we didn't apply $\gamma_\theta$ during sampling?
> > This would be an interesting discussion or ablation.
> > Moreover, in the case without learnable $\gamma_\theta$, i.e.\ when using only the Haar measure, I would expect that the model would still perform fine even if no $R$ is applied during sampling.}
> >
> > The intuition is that we want a diffusion model that is equivariant and to achieve this via stochastic symmetrisation (Cornish, 2024), we require the use of $\gamma_\theta$ within the reverse kernels of our model (Proposition 1). If this was not applied, our model would no longer be guaranteed to be equivariant.
> >
> > We also agree that the case where $\gamma_\theta$ is not applied during sampling is an interesting ablation. We will run this and report back.
> >
> > Further, when $\gamma_\theta$ is the Haar measure, you are correct that the model would be expected to still perform fine even if no $R$ is applied during sampling. In fact, this ablation is exactly the case considered in the comparison between DiT-Aug and SymDiff-H as taking $\gamma_\theta$ to be the Haar measure during training and not using $\gamma_\theta$ during sampling corresponds to data augmentation (with the Haar). See below for further details
> >
> > **Relation to data augmentation**
> >
> > > However, in all experiments the performance of DiT-Aug and SymDiff-H is extremely close together. This makes me believe the application of the augmentation during sampling isn't necessary for SymDiff-H (also see point above), and that SymDiff-H really just corresponds to standard data augmentation.
> >
> > Yes, this is correct -- the behavior of the two models is very similar.
> > At training time, both models are in fact equivalent by Proposition 3.
> > As a result, at test time, we expect the backbone $\epsilon_\theta$ of SymDiff-H to closely match the $\epsilon_\theta$ network learned for the (unsymmetrised) DiT-Aug, which is in turn approximately equivariant. This results in the similar behavior of both models. The purpose of this comparison was not to advocate for the use of SymDiff-H but to provide a natural ablation exploring the case where $\gamma_\theta$ is taken to be the Haar.

---

> ### Author Response · Authors · 2024-11-19
> **Continued response to reviewer u6Xc**
>
> > Related to that, it is not sufficiently well discussed, explained and analyzed how exactly this situation qualitatively changes when is learnt (see point above).
>
> Please see our discussion of the distributions that $\gamma_\theta$ learns above.
>
> **Experimental results**
>
> > Related to the last point above, overall, it seems that SymDiff can be seen as a sophisticated data augmentation strategy (as also pointed out by the authors).
>
> As mentioned above, we would view this as (in a way) the other way around -- data augmentation is a special case of SymDiff. For example, Proposition 3 shows that data augmentation corresponds to SymDiff with the choice of $\gamma_\theta$ being the Haar measure, where we do not only transform the input but also transform the output with $\gamma_\theta$.
>
> > Further, the authors make the point that the comparison to EDM is fair. However, the very different network sizes still make a fair comparison somewhat difficult.
>
> Please see our overall response to all reviewers, which responds to this point.
>
> **Miscellaneous question**
>
> > In 4.2, the authors write that, given the size of GEOM-Drugs, they train a smaller model than on QM9.
> But the GEOM-Drugs dataset is larger and more complex than QM9.
> Wouldn't one want to train a larger, more expressive model in that case, and not a smaller one?
>
> Yes, you are right that it would make sense to train a larger model for a more complex dataset.
> The reason for our choice was that we wanted to match [2] which trains a smaller model on GEOM-Drugs.
> Their motivation for training a smaller network was due to computational limitations, as they did not have enough compute to train a larger EDM model on GEOM-Drugs.
>
> We have now ran our SymDiff model from the QM9 dataset on GEOM-Drugs. We include the updated results below and have also now included these results in the manuscript (Table 4).
>
> | Method | NLL $\downarrow$ | Atm. stability (\%) $\uparrow$ | Val. (\%) $\uparrow$ |
> | ---------- | ------------------------- | ---------------------------------------- | --------------------------- |
> | GeoLDM | --  | 84.4 | 99.3|
> | END | -- | 87.8$\pm{0.99}$ | 92.9$\pm{0.3}$ |
> | EDM | -137.1 | 81.3 |  -- |
> | SymDiff | -301.21$\pm{0.53}$ | 86.16$\pm{0.05}$ | 99.27$\pm{0.1}$ |
> | Data | -- | 86.50 | 99.9
>
> [1] Villar, Soledad, et al. "Scalars are universal: Equivariant machine learning, structured like classical physics." Advances in Neural Information Processing Systems 34 (2021).
>
> [2] Hoogeboom, Emiel, et al. "Equivariant diffusion for molecule generation in 3d." International conference on machine learning (2022)

---

> ### Comment · Reviewer_u6Xc · 2024-11-25
> **Response to Rebuttal**
>
> I would like to thank the authors for the comprehensive reply to my review and I appreciate the additional explanations and results. Some of my concerns have been addressed.
>
> The authors promised to run additional experiments and to update their results:
> > We do not have the final trained model yet, but we will update this before the deadline.
>
> as well as
>
> >We also agree that the case where $\gamma_\theta$ is not applied during sampling is an interesting ablation. We will run this and report back.
>
> I think these experiments would be helpful and insightful and I would like to kindly follow up whether the authors were able to run these experiments.
>
> I also have one rather high-level follow-up question. I believe I now understand the framework reasonably well mathematically, and I agree that the framework can be seen as a generalization of regular data augmentation. Formally, the approach can be considered strictly more flexible. However, I still lack some intuitions: Why exactly would it ever be better to use such "learnt data augmentation" instead of augmenting with random rotations (or, equivalently, the Haar measure)? Intuitively, when being confronted with a problem exhibiting such rotation symmetry, one would assume that random rotation augmentation is exactly the right type of augmentation. Why should there be a preferential direction? However, the presented work does seem to improve upon this. Hence, what quality does the data distribution need to have such that such a more complex augmentation via a more complex $\gamma_\theta$ is beneficial? And, on the contrary, for what sort of data distribution would you expect that the ideal $\gamma_\theta$ is simply the random Haar measure, without refinement? Could you provide some examples and more intuitions?

---

> > ### Author Response · Authors · 2024-11-28
> > **Response to reviewer u6Xc**
> >
> > Thank you for your response; we are glad that we were able to address some of your concerns. We address your remaining concerns below.
> >
> > \
> > **When is a learned $\gamma_\theta$ preferred?**
> >
> > > Why exactly would it ever be better to use such "learnt data augmentation" instead of augmenting with random rotations (or, equivalently, the Haar measure)?
> >
> > Intuition about the utility of a learned $\gamma_\theta$ can be transferred from the existing literature on symmetrising deterministic functions [5,1,4],
> > as opposed to equivariant Markov kernels which is the more general setting that our paper considers.
> > These deterministic methods also involve learning a component analogous to our $\gamma$, and obtain various benefits.
> > For example, the method of [4] constructs a deterministic $G$-equivariant function $\phi_{\theta}$ by computing
> >
> > $$
> > \phi_{\theta}(x) = \int g\cdot f_\theta(g^{-1}\cdot x) \gamma_\theta(dg|x),
> > $$
> >
> > where $f_\theta : X \to \mathbb{R}^d$ is an unconstrained function and $\gamma_\theta(dg|x)$ is a $G$-equivariant Markov kernel. (See also the end of Section 5 of Cornish (2024).)
> >
> > Now, for the choice of $\gamma_\theta(dg|x)$ we have two extreme cases.
> > One takes $\gamma_\theta(dg|x)$ to be a deterministic $G$-equivariant function.
> > This is the approach of [1] called "canonicalization", which maps the input $x$ to a "canonical'" pose due to the fact that $\gamma_\theta$ is equivariant (see Figure 1 of [1]).
> >
> > The benefit reported in [1] is that this allows the prediction problem (which is handled by $k_\theta$) to be decoupled from the problem of ensuring equivariance (which is handled separately by $\gamma_\theta$), leading to better reported performance in many cases.
> >
> > However, an issue with this approach is that [2] proves for many groups of interest (such as the orthogonal group), no continuous canonicalisation function $\gamma$ can exist.
> >
> > The other extreme instead takes $\gamma_\theta(dg|x)$ to be the Haar measure $\lambda(dg)$ (when $G$ is compact) as a general purpose solution. This avoids the pathologies of canonicalization due to its probabilistic nature but doesn't facilitate the decoupling of the learning problem.
> > However, it does encourage the backbone $k_\theta$ itself (and therefore the overall model $\phi_\theta$) to become itself approximately equivariant.
> > In addition, here the additional parameters of $\gamma_\theta$ may also allow $k_\theta$ to discover a better optimum than would be found otherwise, due to the potential benefits of overparameterisation on neural network training [3].
> >
> > From the above discussion, we can see that the benefit of letting $\gamma_\theta$ be a flexible learnable $G$-equivariant Markov kernel is that it allows the overall model to interpolate between these extreme cases (while maintaining equivariance), allowing the optimization process to discover a better minima than either case might access by itself.
> > These points provide the motivation behind [4] in the deterministic setting, which presents improved empirical results in the deterministic setting from using such a $\gamma_\theta$, thus helping to validate this intuition.
> >
> > In our paper, we note that while we are symmetrising Markov kernels rather than deterministic functions, the above intuition still transfers to our setting due to the analogous roles which $k_\theta$ and $\gamma_\theta$ play in our framework. Hence, by allowing $\gamma_\theta$ to be learned, we intuitively obtain the same optimization benefits as reported in [5, 1, 4], but now in the novel context of generative modelling.
> >
> >
> > > Hence, what quality does the data distribution need to have such that such a more complex augmentation via a more complex $\gamma_\theta$ is beneficial?
> >
> > The benefits of our approach are not so much consequences of specific properties of the data distribution (apart from its underlying symmetries) as they are consequences of the model itself.
> > As discussed above, the utility of a learnable $\gamma_\theta$ is in giving the model additional flexibility, which lets it select potentially very complex behaviours according to the specifics of each data distribution it is trained on.

---

> ### Author Response · Authors · 2024-11-28
> **Response to reviewer u6Xc**
>
> **Ablation with $\gamma_\theta$:**
>
> Here are the results of an ablation where we evaluated our SymDiff model three times where we now did not apply $\gamma_\theta$ at sampling time (SymDiff-GammaOff). Please find the results in the table below.
>
> | Method | NLL $\downarrow$ | Atm. stability (\%) $\uparrow$ | Mol. stability (\%) $\uparrow$ | Val. (\%) $\uparrow$ | Uniq. (\%) $\uparrow$
> | ---------- | ------------------------- | ---------------------------------------- | --------------------------- | --------------------------- | --------------------------- |
> | SymDiff | -129.35$\pm$1.07 | 98.74$\pm$0.03 | 87.49$\pm$0.23 | 95.75$\pm$0.10 | 97.89$\pm$0.26
> | SymDiff-GammaOff | -129.40$\pm$1.90 | 98.72$\pm$0.04 | 87.00$\pm$0.24 | 95.43$\pm$0.06 | 97.91$\pm$0.05 |
>
> From the above, we see that not applying $\gamma_\theta$ during sampling causes only a small decrease in performance. This observation suggests that our SymDiff model has learnt an approximately equivariant $k_\theta$ in this case. This is in line with our earlier explanations.
>
> \
> **Distribution of $\gamma_\theta$:**
>
> Please find an image of the distribution of samples from $\gamma_\theta$ from the model described earlier, which we call **SymDiff-Scalars**, [here](https://imgur.com/a/Ig8rKEC).
> We have now trained this model to completion, and the performance metrics for this model are reported in the table below.
>
> | Method | NLL $\downarrow$ | Atm. stability (\%) $\uparrow$ | Mol. stability (\%) $\uparrow$ | Val. (\%) $\uparrow$ | Uniq. (\%) $\uparrow$
> | ---------- | ------------------------- | ---------------------------------------- | --------------------------- | --------------------------- | --------------------------- |
> | SymDiff-Scalars | -129.84$\pm$0.68 | 98.51$\pm$0.01 | 84.37$\pm$0.15 | 94.96$\pm$0.11 | 98.05$\pm$0.29
> | EDM | -110.70$\pm$1.50  | 98.70$\pm$0.10 | 82.00$\pm$0.40 | 91.90$\pm$0.50 | 90.70$\pm$0.60
>
>
> From the above, we observe that SymDiff-Scalars overall performed worse than SymDiff, although it still outperformed EDM. Additionally, we see that the distribution of $\gamma_\theta$ has a much tighter spread than the one by SymDiff. These results further support our earlier discussion, showing that learning more complex $\gamma_\theta$ can also be beneficial.
>
> We hope that these responses address your outstanding concerns (please let us know if not), and if so that you will consider increasing your score.
>
>
> [1] Kaba, Sékou-Oumar, et al. "Equivariance with learned canonicalization functions." International Conference on Machine Learning. PMLR, 2023.
>
> [2] Dym, Nadav, Hannah Lawrence, and Jonathan W. Siegel. "Equivariant frames and the impossibility of continuous canonicalization." arXiv preprint arXiv:2402.16077 (2024).
>
> [3] Allen-Zhu, Zeyuan, Yuanzhi Li, and Yingyu Liang. "Learning and generalization in overparameterized neural networks, going beyond two layers." Advances in neural information processing systems 32 (2019).
>
> [4] Kim, Jinwoo, et al. "Learning probabilistic symmetrization for architecture agnostic equivariance." Advances in Neural Information Processing Systems 36 (2023)
>
> [5] Puny, Omri, et al. "Frame averaging for invariant and equivariant network design." arXiv preprint arXiv:2110.03336 (2021).

---

> ### Comment · Reviewer_u6Xc · 2024-11-30
> **Reviewer response**
>
> I would like to thank the authors for their additional explanations and experimental results. I think these results are indeed helpful to provide context and understanding. I would like to ask the authors to add these experiments as well as the explanation that considers the two extreme cases of $\gamma_\theta$ (I found this very helpful to get some intuitions) to the final paper, for instance in the appendix if space is a concern.
>
> I am willing to raise my score and support publication of the work.

---

> > ### Author Response · Authors · 2024-11-30
> > **Response to reviewer u6Xc**
> >
> > Thank you for your thoughtful and encouraging feedback as well as the support for our work. We appreciate the time and effort you have taken to help improve our work.
> >
> > In response to your suggestion, we will include the requested experiments and the explanation considering the two extreme cases in the final version of the paper.

---

### Author Response · Authors · 2024-11-19
**Overall response to all reviewers**

Thanks very much to all reviewers for your helpful responses, all of which we will try to incorporate.
A general point from all reviewers was around our benchmarking methodology.
Reviewers noted that our SymDiff models had more parameters than our baselines, and so may have had an unfair advantage. We address this here.

The authors are indeed correct that our models have a higher parameter count than our baselines. However, as we are comparing very different architectures, we believe that the number of parameters is not as important as compute time and memory usage.

More specifically, GeoLDM, END and EDM all make use of GNNs. These have relatively few parameters, but rely on message passing, which is very costly in terms of both compute time and memory. In contrast, our SymDiff models only used Transformers, which scale much better, and so can obtain the same compute time and memory performance even with many more parameters.

We attempted to show this with Table 2 and 3 of the paper, but on rereading these we think these results could be considerably streamlined.
Instead, please see the table below, which we have also updated the manuscript to include too (Table 2).

| Method    | Mol. Stability | Sampling (s) | vRAM (GB) | Sec./epoch | # Parameters |
| -------- | ------- | -------- | -------- | -------- | -------- |
| GeoLDM  | $89.40\pm 0.50$  |  0.26 | 27 | 210.93 | 11.4M |
| MuDiff | $89.90\pm1.10$    |  0.89 | 36 | 230.87 | 9.7M |
| END    | $89.10\pm0.10$    | ? | ? | ? | 9.4M |
| EDM |  $82.00 \pm 0.40$|  0.27 | 14 | 88.80 | 5.4M |
| SymDiff* | $89.65\pm0.10$ | 0.21 | 16 | 53.40 | 117.8M |
| SymDiff | $87.49\pm 0.23$  | 0.09 | 7 | 27.20 | 31.2M |

This table shows our QM9 results from Table 1 of the paper, but where we have now included the computational cost of each method.
For brevity, we have included only molecular stability which is often considered as the main performance metric for molecular generation.
As the reviewers requested, we have now also included memory usage (vRAM). Additionally, as the different methods used different batch sizes, we have now replaced the number of iterations per second with seconds per epoch. All the computations were done using a NVIDIA H100 80GB GPU. We excluded END as the code for this method is not publicly available.

Overall, we observe the following:

* Despite having many more parameters, even our very large SymDiff* model outperformed all other models in terms of sampling time, vRAM, and seconds/epoch (apart from slightly more vRAM compared with EDM), and did on par or better in terms of molecular stability.

* Our smaller SymDiff model reduced these costs considerably further, and still remained competitive in terms of molecular stability. Compared with EDM, which is the most directly comparable baseline, it performed significantly better.

In other words, SymDiff methods achieve the same or better model performance at a considerably lower computational budget.

In addition, the flexibility of SymDiff means it could also be used in conjunction with ideas from these other methods as well.
For example, MuDiff uses a more complex molecular model that also uses information about molecular bond types.
The same approach could be used with SymDiff, and it seems reasonable to expect performance benefits there.
Similar arguments also hold for the autoencoder/latent encoding of GeoLDM, and the learned forward process of END.

Finally, although it is not directly related to computational efficiency, we also note that SymDiff is much simpler to implement compared with these other baselines.
Algorithms 1 and 2 of our paper, which are both only a few lines of code, give a complete picture of our framework, where the networks involved ($f_\theta$ and $\epsilon_\theta$) can be very simple and scalable off-the-shelf models (like DiTs).
In contrast, these other baselines require a much more involved implementation process to ensure the desired equivariant conditions hold (see for example Appendix A.3 of the MuDiff paper).
Given that we also obtain overall benefits computationally, this seems to be a significant ``win-win'', especially since considerable attention has been devoted to the problem of obtaining equivariance to-date.


P.S.: We realised a mistake in the times reported in this responses and have now fixed these in both the responses and manuscript.

---

### Meta-Review · Area_Chair_bsT5 · 2024-12-17

**Metareview:**

This paper introduces SymDiff, a novel method for constructing equivariant diffusion models through stochastic symmetrization. Unlike prior approaches that rely on intrinsically equivariant neural networks, SymDiff offers greater flexibility for geometric data. Experiments on the QM9 and GEOM-Drugs datasets demonstrate that SymDiff outperforms EDM and achieves competitive performance with other baselines, while offering improved scalability. Conceptually, SymDiff generalizes data augmentation by learning and applying a stochastic augmentation process that enforces equivariance.

All reviewers agree that the work is solid and interesting.

**Additional Comments On Reviewer Discussion:**

The authors have addressed most reviewers' questions with extra experimental evaluations.

---

### Decision · Program_Chairs · 2025-01-22

Accept (Poster)